# SYNCOGEN: SYNTHESIZABLE 3D MOLECULE GENERATION VIA JOINT REACTION AND COORDINATE MODELING

**Andrei Rekesh**[1,2,*] **Miruna Cretu**[3], **Dmytro Shevchuk**[1,2], **Vignesh Ram Somnath**[4],
**Pietro Liò**[3], **Robert A. Batey**[1], **Mike Tyers**[1,2], **Michał Koziarski**[1,2,5], **Cheng-Hao Liu**[6,7,8*]

[1]University of Toronto, [2]The Hospital for Sick Children, [3]University of Cambridge,
[4]ETH Zürich, [5]Vector Institute, [6]Mila – Quebec AI Institute, [7]McGill University, [8]Caltech

## ABSTRACT

Synthesizability remains a critical bottleneck in generative molecular design. While recent advances have addressed synthesizability in 2D graphs, extending these constraints to 3D for geometry-based conditional generation remains largely unexplored. In this work, we present SYNCOGEN (Synthesizable Co-Generation), a single framework that combines simultaneous masked graph diffusion and flow matching for synthesizable 3D molecule generation. SYNCOGEN samples from the joint distribution of molecular building blocks, chemical reactions, and atomic coordinates. To train the model, we curated SYNSPACE, a dataset family containing over 1.2M synthesis-aware building block graphs and 7.5M conformers. SYNCOGEN achieves state-of-the-art performance in unconditional small molecule graph and conformer co-generation. For protein ligand generation in drug discovery, the amortized model delivers superior performance in both molecular linker design and pharmacophore-conditioned generation across diverse targets without relying on any scoring functions. Overall, this multimodal non-autoregressive formulation represents a foundation for a range of molecular design applications, including analog expansion, lead optimization, and direct *de novo* design.

## 1 INTRODUCTION

Generative models significantly enhance the efficiency of chemical space exploration in drug discovery by directly sampling molecules with desired properties. However, a key bottleneck in their practical deployment is low synthetic accessibility; that is, generated molecules are often difficult or impossible to produce in the laboratory (Gao & Coley, 2020). To address this limitation, recent work has turned to template-based methods that emulate the chemical synthesis process by constructing synthesis trees that link molecular building blocks through known reaction templates (Koziarski et al., 2024; Cretu et al., 2024; Seo et al., 2024; Gaiński et al., 2025; Gao et al., 2024; Jocys et al., 2024; Swanson et al., 2024). These representations, while useful for downstream experimental validation, do not describe the underlying 3D geometry and thus cannot capitalize on the conformational information that is often crucial for diverse chemical and biological properties.

Parallel advances in generative molecular design have explored spatial modeling at the atomic level. Inspired by advances in protein structure prediction (Yang et al., 2025; Campbell et al., 2024; Wang et al., 2025) and the development of generative frameworks such as diffusion and flow matching, recent work has focused on directly sampling 3D atomic coordinates of small molecules (Hassan et al., 2024; Jing et al., 2022; Fan et al., 2024). These methods learn to generate spatially meaningful, property-aligned conformations along with molecular graphs. The ability to model atomic coordinates directly increases the expressivity of these approaches, enabling applications such as pocket-conditioned generation (Lee & Cho, 2024), scaffold hopping (Torge et al., 2023; Yoo et al., 2024), analog discovery (Sun et al., 2025), and molecular optimization (Morehead & Cheng, 2024). However, without considering practical synthesis routes, integrating synthesizability constraints into these models remains a major challenge, and most existing 3D generative approaches do not ensure that proposed molecules can be made in practice.

---

*Correspondence to `a.rekesh@mail.utoronto.ca` and `chenghao.liu@mail.mcgill.ca`

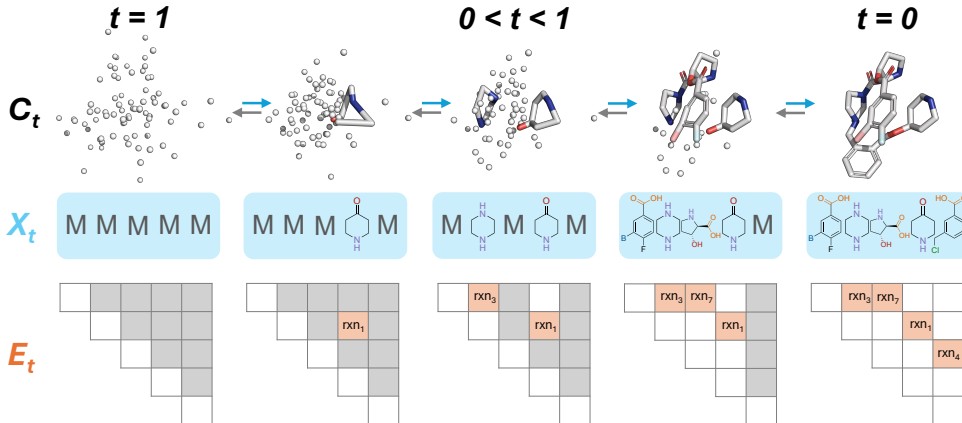

Figure 1: SYNCOGEN is a simultaneous masked graph diffusion and flow matching model that generates synthesizable molecules in 3D coordinate space. Each node corresponds to a building block, and edges encode chemical reactions. Note that graphs are not necessarily path graphs, the leaving groups are not displayed, and there is no order to which nodes and edges are denoised.

This work introduces SYNCOGEN (Synthesizable Co-Generation), a generative modeling framework aiming to bridge the gap between 3D molecular generation and practical synthetic accessibility (Figure 1). Our main contributions are as follows:

- **Generative Framework:** We propose a novel generative framework that combines masked graph diffusion with flow matching in unified time to jointly sample from the distribution over building block reaction graphs and of 3D coordinates, tying structure- and synthesis-aware modeling.

- **Molecular Dataset:** We curate a new dataset family SYNSPACE, comprising 1.2M synthesizable molecules represented as building block reaction graphs, along with 7.5M associated low-energy conformations. Compared to synthon-based datasets, SYNSPACE enables models to generate more readily synthesizable molecules and directly suggest streamlined synthetic routes.

- **Empirical Validation:** We demonstrate that SYNCOGEN achieves state-of-the-art performance in 3D molecule generation, producing physically realistic conformers while explicitly tracing reaction steps. Ablations show our modelling choices are crucial for the performance. Importantly, SYNCOGEN performs 3D conditional molecular generation tasks including linker design and pharmacophore-conditioned generation, highlighting its applicability for drug discovery.

Our code can be found at `https://github.com/andreirekesh/SynCoGen`. Our data can be found at `https://huggingface.co/datasets/DreiSSB/SynSpace`.

## 2 BACKGROUND AND RELATED WORK

**Flow Matching.** Given two distributions $\rho_0$ and $\rho_1$, and an interpolating probability path $\rho_t$ such that $\rho_{t=0} = \rho_0$ and $\rho_{t=1} = \rho_1$, flow matching (Lipman et al., 2023; Albergo et al., 2023; Liu et al., 2023; Peluchetti, 2023; Tong et al., 2023) aims to learn the underlying vector field $u_t$ that generates $\rho_t$. Since $u_t$ is not known in closed form, flow matching instead defines a conditional probability path $\rho_{t|1}$ and its corresponding vector field $u_{t|1}$. The marginal vector field $u_t$ can then be learnt with a parametric $v_\theta$ by regressing against $u_{t|1}$ with the CFM objective:

$$\mathcal{L}_{\text{CFM}}(\theta) = \mathbb{E}_{t, \mathbf{x}_1 \sim \rho_1, \mathbf{x} \sim \rho_{t|1}(\cdot|x_1)} ||v_t(\mathbf{x}; \theta) - u_{t|1}(\mathbf{x}|\mathbf{x}_1)||^2 \tag{1}$$

**Masked Discrete Diffusion Models.** Let $\mathbf{x} \sim \rho_{\text{data}}$ be a one-hot encoding over $K$ categories. Discrete diffusion models (Austin et al., 2021; Sahoo et al., 2024; Shi et al., 2024) map the complex data distribution $\rho_{\text{data}}$ to a simpler distribution via a Markov process, with absorbing (or masked) diffusion being the most common. In the masked diffusion framework, the forward interpolation process $(\rho_t)_{t\in[0,1]}$ with the associated noise schedule $(\alpha_t)_{t\in[0,1]}$ results in marginals $q(\mathbf{z}_t|\mathbf{x}) = \text{Cat}(\mathbf{z}_t; \alpha_t\mathbf{x} + (1 - \alpha_t)\mathbf{m})$, where $\mathbf{z}_t$ and $\mathbf{m}$ denote intermediate latent variables and the one-hot

encoding for the special [MASK] token, respectively. The posterior can be derived as:

$$q(\mathbf{z}_s|\mathbf{z}_t, \mathbf{x}) = \begin{cases} \text{Cat}(\mathbf{z}_s; \mathbf{z}_t), & \mathbf{z}_t \neq \mathbf{m} \\ \text{Cat}(\mathbf{z}_s; \frac{(1-\alpha_t)\mathbf{m}+(\alpha_s-\alpha_t)\mathbf{x}}{1-\alpha_t}), & \mathbf{z}_t = \mathbf{m} \end{cases} \quad (2)$$

The optimal reverse process $p_\theta(z_s \mid z_t)$ takes the same form but with $x_\theta(z_t, t)$ in place of the true $\mathbf{x}$. We adopt the zero-masking and carry-over unmasking modifications of Sahoo et al. (2024).

**Multimodal Generative Models.** Multimodal data generation (e.g. text-images, audio-vision, sequences/atomic types and 3D structures) represents a challenging frontier for generative models and has seen growing interest in recent times. Current approaches for this task typically either – 1) tokenize multimodal data into discrete tokens, followed by a autoregressive generation (Meta, 2024; Xie et al., 2024; Lu et al., 2024), or 2) utilize diffusion / flow models for each modality in its native space (Lee et al., 2023; Zhang et al., 2024; Campbell et al., 2024; Irwin et al., 2025). Diffusion and flow models also offer flexibility in terms of coupled (Lee et al., 2023; Irwin et al., 2025) or decoupled (Campbell et al., 2024; Bao et al., 2023; Kim et al., 2024) diffusion schedules across modalities. SYNCOGEN uses a coupled diffusion schedule but at two resolutions, with discrete diffusion for graphs of building blocks and reactions, and a flow for atomic coordinates in building blocks.

**3D Molecular Generation.** Several recent works (Irwin et al., 2025; Le et al., 2024; Vignac et al., 2023; Huang et al., 2023; Dunn & Koes, 2024) have studied unconditional molecular structure generation by sampling from the joint distribution over atom types and coordinates. However, these models lack the ability to constrain the design space to synthetically accessible molecules. In concurrent work, (Shen et al., 2025) uses generated 3D structures to guide GFlowNet policies in designing the graph of *synthon*-based linear molecules, but does not account for structural quality.

**Synthesizable Molecule Generation.** Beyond directly optimizing synthesizability scores (Liu et al., 2022; Guo & Schwaller, 2025) – which are often unreliable – the predominant approach to ensuring synthetic accessibility in generative models is to incorporate reaction templates. Early methods explored autoencoders (Bradshaw et al., 2019; 2020), genetic algorithms (Gao et al., 2022), and reinforcement learning (Gottipati et al., 2020; Horwood & Noutahi, 2020). Recently, GFlowNet-based (Koziarski et al., 2024; Cretu et al., 2024; Seo et al., 2024; Gaiński et al., 2025) and transformer-based (Gao et al., 2024; Jocys et al., 2024) methods have gained prominence. Such generative models have already shown practical utility in biological discovery tasks (Swanson et al., 2024). However, most methods only generate molecular graphs and do not produce 3D structures. The recent CGFlow Shen et al. (2025) performs 3D generation via a GFlowNet policy augmented with flow matching; however, CGFlow optimizes a reward and typically requires a full training for each target pocket.

## 3 DATASET

Training a synthesizability-aware model to co-generate both 2D structures and 3D positions requires a dataset of easily synthesizable molecules in an appropriate format. In addition to atomic coordinates, this includes a graph-based representation from which plausible synthetic pathways can be inferred. A common approach is to use synthons—theoretical structural units that can be combined to form complete molecules(Baker et al., 2024; Grigg et al., 2025; Medel-Lacruz et al., 2025). Synthon-based representations do not guarantee the existence of a valid synthesis route, and they do not directly provide one even if it exists. Moreover, they lack the flexibility to constrain the reaction space, which is often critical when prioritizing high-yield, high-reliability reactions or operating within the limits of automated synthesis platforms such as self-driving labs (Abolhasani & Kumacheva, 2023).

Alternatively, many synthesis-aware generators employ external reaction simulators, such as RDKit, to couple building blocks iteratively. While convenient, such black-box steps offer no fine-grained control when a reagent has multiple *reaction centers*, distinct atoms or atom sets that can each serve as the site of bond formation or cleavage in a reaction. They also do not define atom mappings between reactants and products, making it impossible to trace product atoms back to their parent building blocks, which complicates edge assignment in building block graph generation. To overcome these limitations, we curate a new family of datasets, SYNSPACE (Figure 2), comprising building block-level reaction graphs pairs with corresponding atom- and block-level graphs. We then calculate multiple 3D conformations for each graph using semi-empirical methods (Bannwarth et al., 2019).

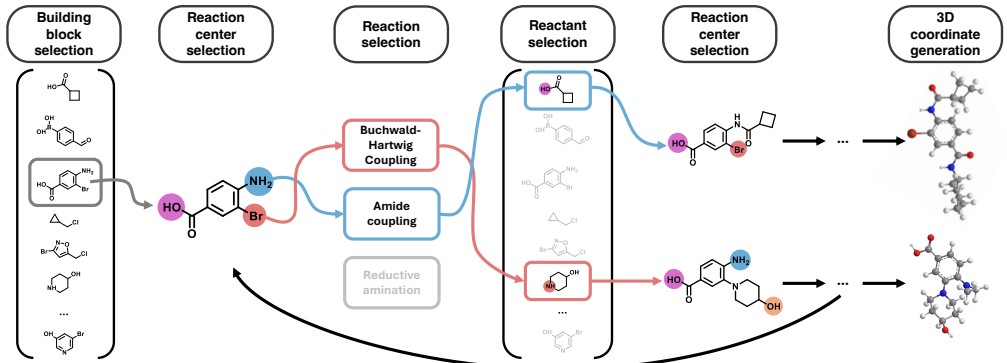

Figure 2: **Overview of SYNSPACE creation process.** Highly synthesizable molecules are procedurally constructed by iteratively sampling synthesis pathways from a set of building blocks and reactions. Starting from an initial block, the procedure selects a reaction center, a compatible reaction, and a suitable reactant. After the final structure is assembled, multiple low-energy 3D conformations are generated. We provide *two* SYNSPACE datasets from two vocabularies, a practically focused core set and an extended variant; each dataset contains 600k graphs with 3-4M conformers.

## 3.1 SYNSPACE: GRAPH GENERATION

We first construct *two* curated vocabularies adapted from the collection proposed by Koziarski et al. (2024). The first vocabulary pairs 93 low-cost, commercially available building blocks with 19 high-yield reaction templates, defining a virtual synthesis space of over a billion molecules. The second vocabulary is a superset with 378 building blocks with 26 reactions, expanding the synthesis space to over a trillion molecules. All building blocks were selected because they are known to undergo the chosen reactions, acknowledging that the presence of a nominally compatible functional group alone does not guarantee participation in the corresponding transformation. We utilize reactions that (1) ensure all product atoms originate from the two input reagents, and (2) involve at most one leaving group per reagent. We emphasize that these constraints yield simple, robust chemistries that are routinely executed and support rapid multi-step synthesis from inexpensive, in-stock reagents.

We procedurally generate SYNSPACE from the smaller vocabulary, or SYNSPACE-L from the larger superset, by iteratively coupling building block graphs at their reaction centers with compatible reaction templates (Appendix A.2). For SYNSPACE, we obtained 622,766 building block reaction graphs, each constructed from 2 to 4 sequential reactions. For each molecule, we generate multiple 3D conformations (Section 3.2), yielding 3,360,908 conformers. Similarly, SYNSPACE-L contains 600,000 graphs and 4,223,367 conformations. Unless otherwise noted, all models are trained on SYNSPACE, which emphasizes practicality as its fewer building blocks are more readily stocked, whereas SYNSPACE-L is reserved for when a larger, more exploratory search space is required.

SYNSPACE contains diverse molecules that are drug-like (e.g., LogP $\sim 2.5$; broad range of topological polar surface areas; large fraction of sp$^3$ carbons). Importantly, compared to Geom-Drugs (Axelrod & Gomez-Bombarelli, 2022), SYNSPACE contains substantially more unique Murcko scaffolds, indicating breadth despite the building block space. With a larger accessible space, SYNSPACE-L preserves similar physicochemical profiles and scaffold diversity. See Appendix A.3 for details.

**Note: Injectivity.** Many commercial building blocks contain multiple reaction centers, each compatible with a different set of corresponding reaction centers on other blocks. Thus, a building block-level reaction graph $G_b = (X, E)$ is not fully specified when edges are parametrized by the reaction alone. To achieve an injective correspondence, we label edges from node $i$ to $j > i$ by the triple $e_{ij} = (r, v_i, v_j)$, where $r$ is the coupling reaction and $(v_i, v_j)$ are the participating reaction centers on the source and destination blocks, respectively. Stereoisomers that form during reactions collapse to the same $(X, E)$ representation, but this granularity suffices for our current scope.

## 3.2 SYNSPACE: CONFORMATION GENERATION

For each molecular graph, 50 initial conformers were generated using the ETKDG (Riniker & Landrum, 2015) algorithm (RDKit implementation). These structures were energy-minimized using the MMFF94 force field, and all conformers within 10 kcal/mol of the global minimum were retained.

The resulting geometries were then re-optimized with the semi-empirical GFN2-xTB (Bannwarth et al., 2019) method, after which the same 10 kcal/mol energy threshold was applied. At every stage, redundant structures were removed by geometry-based clustering (RMSD < 1.5Å). This workflow yields, on average, 5.4 distinct conformers per graph. Relative to exhaustive approaches such as CREST (Pracht et al., 2024), the workflow is several orders of magnitude faster; despite occasionally omitting some conformations, the retained structures are diverse and reproduce the bond-length, bond-angle, and dihedral-angle distributions observed in CREST-derived datasets (see Section 5.1).

## 3.3 SYNSPACE: PHARMACOPHORE GENERATION

For each conformer of a molecule in SYNSPACE and SYNSPACE-L, we generate a pharmacophore profile consisting of one-hot pharmacophore types $X_{\text{pharm}} \in \{0,1\}^{N_{\text{pharm}} \times N_{\text{types}}}$ and positions $C_{\text{pharm}} \in \mathbb{R}^{N_{\text{pharm}} \times 3}$ using ShePhERD Adams et al. (2025). Here, $N_{\text{pharm}}$ and $N_{\text{types}}$ correspond to the number of pharmacophore features and the number of pharmacophore types, respectively.

## 4 METHODS

**Notation.** Let $\mathcal{B}$ be the building-block vocabulary and $\mathcal{R}$ the set of reaction templates, with cardinalities $B := |\mathcal{B}|$ and $R := |\mathcal{R}|$. Let $N$ denote the maximum number of building blocks per molecule and $M$ the maximum number of atoms per building block. For each block $b \in \mathcal{B}$ we denote its set of reaction-center atoms by $\mathcal{V}(b)$; the global maximum of these counts is $V_{\text{max}} := \max_{b \in \mathcal{B}} |\mathcal{V}(b)|$. Hence, tensor shapes contain factors such as $B + 1$ (to accommodate the masked token $\pi_X$ in $X$), $R\,V_{\text{max}}^2 + 2$ (to accommodate the no-edge and masked tokens $\lambda_E$ and $\pi_E$), together with the bounds $N$ and $M$ introduced above.

**SYNCOGEN.** SYNCOGEN generates building block-level reaction graphs and coordinates. Each molecule is represented by a triple $(X, E, C)$ where $X \in \{0,1\}^{N \times |\mathcal{B}|+1}$ encodes the sequence of building-block identities, $E \in \{0,1\}^{N \times N \times |\mathcal{R}|V_{\text{max}}^2 + 2}$ labels the coupling reaction (and centers) between every building block pair, and $C \in \mathbb{R}^{N \times M \times 3}$ stores all atomic coordinates. We detail the parameterization of graphs $(X, E)$ in Appendix B.4. Training combines two diffusion schemes: 1) a **discrete absorbing process** on $(X, E)$ using the categorical forward kernel of Sahoo et al. (2024), and 2) a **continuous, visibility-aware process** on $C$ whose endpoints are (i) a rototranslationally-aligned isotropic Gaussian and (ii) a re-centered ground truth, considering all "visible" atoms in the prior (see Section 4.2). For an intuitive schematic and description of the training procedure, see Appendix B.1.

### 4.1 MODEL ARCHITECTURE

We adapt a $SE(3)$-equivariant architecture originally designed for all-atom molecular design (SEMLAFLOW (Irwin et al., 2025)) as the principal backbone to generate both coordinates and graphs. At each timestep $t$, SYNCOGEN predicts building block logits $L_t^X, L_t^E$ and a shifted coordinate estimate $\hat{\tilde{C}}_0^t$. The loss is the weighted sum of the cross-entropy term $\mathcal{L}_{\text{graph}}$ on $(X, E)$, the masked coordinate MSE term $\mathcal{L}_{\text{MSE}}$, and the short-range pairwise distance term $\mathcal{L}_{\text{pair}}$ (see Appendices B.7 and B.16 for details). We define additional building-block-to-atom featurization in Appendix B.5 and atom-to-building-block output layers in Appendix B.10.

**Pharmacophore Conditioning Backbone.** To accommodate pharmacophores as conditioning information, we design a modified backbone to represent each as an "atom" with no weight during centering operations. After atom featurization, pharmacophore types are fed through a separate featurization head and concatenated to invariant atom type features, i.e. $X_{\text{model}} = [\text{MLP}_{\text{atom}}(X_{\text{atom}}), \text{MLP}_{\text{pharm}}(X_{\text{pharm}})] \in \mathbb{R}^{(N+N_{\text{pharm}}) \times d_x}$. Pharmacophore coordinates are concatenated directly to atomic coordinates, $C_{\text{model}} = [C, C_{\text{pharm}}] \in \mathbb{R}^{(N+N_{\text{pharm}}) \times 3}$, and therefore undergo identical data augmentation beforehand (including that induced by data pairing, see Section 4.2). $C_{\text{model}}$ and $X_{\text{model}}$ are then passed to the equivariant-invariant dynamics module. Prior to final output layers, expanded atom-level hidden-layer outputs are truncated to the total number of atoms $NM$.

### 4.2 NOISING SCHEMES

**Graph Noising.** We noise true graphs $(X_0, E_0)$ to obtain $(X_t, E_t)$ using the procedure described in Section 2. In practice, as all true edge matrices $E_0$ are symmetric, we symmetrize the sampled probabilities for the noising and denoising of $E_t$ correspondingly (see Appendix B.11).

**Coordinate Noising**   During sampling, for any time $t$ where some $X_t$ contains a masked building block, we do not know the block's identity or atom count and thus represent its coordinates by a vector containing $M$ atoms of unknown type, where $M$ is a chosen upper bound on the number of atoms in a building block. To match this lack of information at training time, we perform the following: (i) First, we generate a noised graph $(X_t, E_t)$ and draw $C_1 \sim \mathcal{N}(0, I)^{3 \times (NM)}$. (ii) We then design a *visibility mask* $S_t$ that considers all $M$ atoms for each noised building block containing $m \le M$ atoms in $X_t$ as valid. (iii) To keep atom counts identical within individual data pairs, $S_t$ is applied to both $C_0$ and $C_1$. (iv) The additional $M - m$ "padding" atoms in $C_1$ are copied to $C_0$ to create a modified ground-truth $\tilde{C}_0$. (v) With a consistent number of atoms in place, both are centered. For a visual diagram describing this procedure, see Appendix B.1.

Thus, we construct centered, visibility-masked data-noise coordinate pairs $(\tilde{C}_1, \tilde{C}_0)$ that both contain $|S_t|$ "visible" atoms to match the information available to the model during sampling. Input to the model $C_t$ is then obtained by linearly interpolating $C_t = (1 - t)\tilde{C}_0 + t(\tilde{C}_1)$. Essentially, we task the model with rearranging the true atoms while disregarding padding by learning to fix padding atoms in place. See Algorithm 2 for formalization. We note a caveat in equivariance in Appendix B.6.

**Flexible Atom Count.**   Most 3D molecule generation methods require specifying the number of atoms during inference. Because the prior of SYNCOGEN is over building blocks, we naturally handle a flexible number of atoms during generation and model any excessive atoms as padding.

### 4.3   TRAINING-TIME CONSTRAINTS

For discrete diffusion, SYNCOGEN inherits training-time simplifications from MDLM (Sahoo et al., 2024), including zero masked logit probabilities and carry-over logit unmasking during sampling. In addition, we implement the following:

1. **No-Edge Diagonals.** We set the diagonals of all edge logit predictions $L_\theta^E$ to no-edge, as no building block has a coupling reaction-induced bond to itself.

2. **Edge Count Limit.** Let $k_t := \sum_{1 \le i < j \le n} \mathbb{1}(E_t[i, j, \cdot] \notin \{\pi_E, \lambda_E\})$ be the number of unmasked true edges in the upper triangle of $E_t$. If $k_t = n - 1$, we have the correct number of edges for a molecule containing $n$ building blocks and therefore set all remaining edge logits to $\lambda_E$.

3. **Compatibility Masking.** Assume that for some $E_t$ an edge entry is already denoised, $E_t[i, j, \cdot] = (r, v_i, v_j)$, meaning that building block $i$ reacts with building block $j$ via reaction $r$ and centers $v_i \in \mathcal{V}(X_i)$, $v_j \in \mathcal{V}(X_j)$. Define the sets of *center-matched reagents*

$$\begin{aligned} \mathcal{B}_{r,v}^A &:= \{\, b \in \mathcal{B} \mid (b, v) \text{ matches reagent A in } r \,\}, \\ \mathcal{B}_{r,v}^B &:= \{\, b \in \mathcal{B} \mid (b, v) \text{ matches reagent B in } r \,\}. \end{aligned} \tag{3}$$

For every node slot $i$ (resp. $j$) we construct a $|\mathcal{B}|$-dimensional binary mask

$$\mathcal{X}_{i,k} = \mathbb{1}[b_k \in \mathcal{B}_{r,v_i}^A], \mathcal{X}_{j,k} = \mathbb{1}[b_k \in \mathcal{B}_{r,v_j}^B], k = 1, \dots, |\mathcal{B}|. \tag{4}$$

so that the soft-max for $X_t[i, \cdot]$ (resp. $X_t[j, \cdot]$) is evaluated only over the 1-entries of $\mathcal{X}_i$ (resp. $\mathcal{X}_j$). Analogously, once a node identity $X_t[j] = b$ is denoised, incoming edge channels $(i, j)$ with $j > i$ are masked to reactions $e = (r, v_i, v_j)$ such that $b \in \mathcal{B}_{r,v_i}^B$.

For a visual diagram of the above, see Appendix B.2. Put simply, we restrict logits to disallow loops (e.g. macrocycles, which are often synthetically challenging), to impose a limit on the number of edges, and to better ensure the selection of chemically compatible building blocks and reactions.

### 4.4   SAMPLING

Sampling begins by drawing a building block count $n \sim \mathrm{Cat}(\pi_{\text{frag}})$, setting the node and edge tensors to the masked tokens, $X_1[i, \cdot] = \pi_X$, $E_1[i, j, \cdot] = \pi_E$ for every $0 \le i, j < N$, and padding all $(i \ge n)$ rows/columns with the no–edge token $\lambda_E$. The initial coordinates are an isotropic Gaussian $C_1 \sim \mathcal{N}(0, I)^{N \times M \times 3}$. From this state, each step (i) recenters the current coordinates by the visibility mask $S_t$ derived from $X_t$, (ii) generates node and edge logits and coordinate predictions with the trained model, (iii) draws the next discrete state from (ii), and (iv) updates coordinates via an Euler step. After a final, deterministic pass, we calculate $(\hat{X}_0, \hat{E}_0) = \arg\max_k L_\theta^E[\cdots, k]$ and center the coordinates to yield the molecule $(\hat{X}_0, \hat{E}_0, \hat{C}_0)$. Complete pseudocode is provided in Appendix B.8.

We note our discrete and continuous schemes share a unified time. Lastly, we find inference annealing on the coordinates (see Appendix D.2) yields small performance gains at sampling time.

**Note: Inference-Time Edge Constraints.** By construction, a molecule containing $n$ connected building blocks contains exactly $n - 1$ edges, and building block $j > 0$ has a single unique parent $i < j$. Consequently, sampling of redundant or impossible edges can be eliminated at inference time as described in Appendix B.9 and visualized in Appendix B.3.

## 5 EXPERIMENTS

### 5.1 *De Novo* 3D MOLECULE GENERATION

We first study SYNCOGEN in unconditional molecule generation jointly with 3D coordinates and reaction graphs. We evaluate SYNCOGEN against several recently published all-atom generation frameworks which produce 3D coordinates, including SemlaFlow (Irwin et al., 2025), EQGAT-Diff (Le et al., 2024), MiDi (Vignac et al., 2023), JODO (Huang et al., 2023), and FlowMol (Dunn & Koes, 2024). To isolate modeling effects from data, we retrain SemlaFlow on atomic types/coordinates in SYNSPACE for the same number of epochs as SYNCOGEN.

For each model, we sample 1000 molecules and compute stringent metrics capturing chemical soundness, synthetic accessibility, conformer quality, and distributional fidelity. Regarding the molecular graphs, we report the RDKit sanitization validity (Valid.) and retrosynthetic solve rate (AiZynthFinder (Genheden et al., 2020) (AiZyn.) and Syntheseus (Maziarz et al., 2025) (Synth.)). For conformers, we introduce two physics-based metrics: the median non-bonded interaction energies per atom via the forcefield method GFN-FF and via the semiempirical quantum chemistry method GFN2-xTB Bannwarth et al. (2019); Spicher & Grimme (2020); we also check PoseBusters (Buttenschoen et al., 2024) validity rate (PB). We evaluate the diversity (Div.) as the average pairwise Tanimoto dissimilarity of the Morgan2 fingerprints, novelty (Nov.) as the percentage of candidates not appearing in the training set, and the Fréchet ChemNet Distance (Preuer et al., 2018) (FCD) between generated samples and the training distribution. See Appendix D.4 for details.

Table 1: **Comparison of generative methods for *de novo* 3D molecule generation.**

| *Group* | **Method** | **Primary metrics** | | | | | **Secondary metrics** | | | |
|---|---|---|---|---|---|---|---|---|---|---|
| | | Valid. ↑ | AiZyn. ↑ | Synth. ↑ | GFN-FF ↓ | xTB ↓ | PB ↑ | FCD ↓ | Div. ↑ | Nov. ↑ |
| *Rxns & coords* | SYNCOGEN | **96.7** | **50** | **72** | **3.01** | **-0.91** | **87.2** | **2.91** | 0.78 | 93.9 |
| *Atoms & coords* | SEMLAFLOW | 93.3 | 38 | 36 | 5.96 | -0.72 | **87.2** | 7.21 | 0.85 | 99.6 |
| | SEMLAFLOW SYNSPACE | 72.0 | 27 | 48 | 3.27 | -0.80 | 60.3 | 2.95 | 0.80 | 93.0 |
| | EQGAT-diff | 85.9 | 37 | 24 | 4.89 | -0.73 | 78.9 | 6.75 | **0.86** | 99.5 |
| | MiDi | 74.4 | 33 | 31 | 4.90 | -0.74 | 63.0 | 6.00 | 0.85 | 99.6 |
| | JODO | 91.1 | 38 | 31 | 4.72 | -0.74 | 84.1 | 4.22 | 0.85 | 99.4 |
| | FlowMol-CTMC | 89.5 | 24 | 25 | 5.91 | -0.68 | 69.3 | 13.0 | **0.86** | **99.8** |
| | FlowMol-Gaussian | 48.3 | 6 | 8 | 4.24 | -0.71 | 30.7 | 21.0 | **0.86** | 99.7 |

See Table 1 for results, and Figures 15 and 18 for examples. For chemical reasonableness, SYNCOGEN generates almost entirely valid molecules. Our generation details the reaction and building blocks in a multi-step reaction pathway, and as a result, our molecules are significantly more synthesizable compared to baseline methods. Because AiZynthFinder and Syntheseus solve only 50–70 % of known drug-like molecules, our 50–72 % scores likely underestimate true synthesizability. A rigorous conformer geometry and energy comparison between all methods is provided in Appendix D.5.

Structurally, the generated conformers reproduce the data energy distributions and have very favorable non-covalent interaction energies as evaluated by semi-empirical quantum-chemistry methods, especially when compared to the baseline methods (Table 1 and Figure 3). This is evident from the lack of structural changes upon further geometric relaxation (Figure 16). The Wasserstein-1 distances and Jensen-Shannon divergence can be found in Appendix D.5 and Figure 14. The low non-bonded energies indicate SYNCOGEN learns to sample many intramolecular interactions (Figure 15). Quantitatively, 87% of these conformers pass PoseBusters pose plausibility checks. Furthermore, SYNCOGEN reproduces the delicate data distribution of bond lengths, angles, and dihedrals (Figures 3

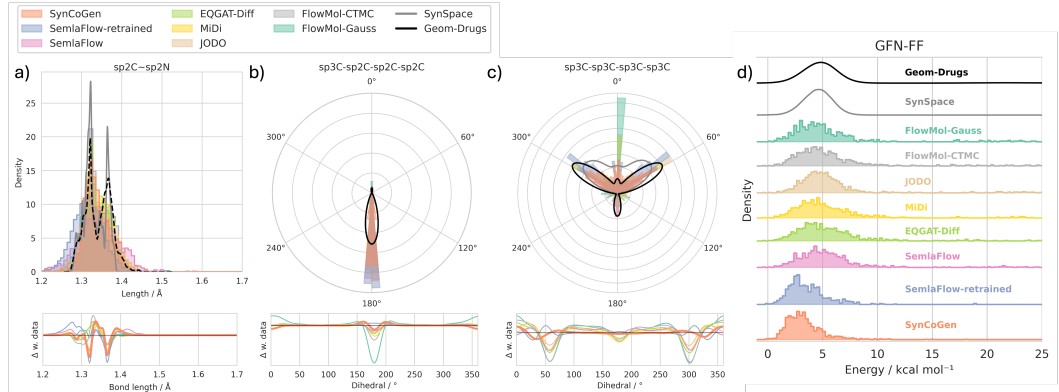

Figure 3: **Conformer geometry and energy distribution.** Distributions of a) bond lengths, b-c) dihedral angles, d) average per-atom GFN-FF non-bonded interaction energies. Solid curves denote training data densities; lower subpanels in (a-c) show deviations between generated samples and data.

and 14). For example, SYNCOGEN generates fewer $sp^2$C-$sp^2$N bonds that are too short, captures sharp bond angle distributions (e.g., $sp^3$C-$sp^3$C-$sp^3$N), and replicates both flexible dihedral angle distribution (e.g. $sp^3$C-$sp^3$C-$sp^3$C-$sp^3$C) and rigid dihedral angles (e.g. $sp^3$C-$sp^2$C-$sp^2$C-$sp^2$C).

Beyond sample quality, SYNCOGEN also captures the training distribution as indicated by the low FCD, while generally producing novel molecules. In exchange for synthesizability, the generated samples have slightly lower diversity due to using a (limited) set of reaction building blocks. All generated samples are unique. Furthermore, the multi-modal model can perform zero-shot conformer generation at a quality similar to ETKDG(RDKit) when given random reaction-graphs (Table 7).

Our various training-time ablations (Table 3) show that the largest performance gains originate from our chemistry-sensitive graph constraints and self-conditioning, with small contributions from other training/sampling details. A large performance gap between SYNCOGEN and SemlaFlow retrained on SYNSPACE further shows that our training procedure, rather than the architecture or dataset, is the primary driver of performance. Appendix D.2 shows sampling-time ablations on schedules, annealing, and edge sampling strategies, which show the joint schedule is beneficial for stable co-generatio.

Finally, we demonstrate that SYNCOGEN is not limited by vocabulary size. When trained on SYNSPACE-L, whose search space is larger by several orders of magnitude (Appendix D.3), the model retains high RDKit validity, realistic conformer energies, and strong retrosynthesis solve rates. This indicates that SYNCOGEN can be readily scaled to broader chemical spaces with little sacrifice on generation quality or synthesizability.

## 5.2 MOLECULAR INPAINTING FOR FRAGMENT LINKING

To demonstrate SYNCOGEN in drug discovery, we study fragment linking (Bancet et al., 2020) to design *easily synthesizable* analogs of hard-to-make drugs. Fragment linking can create potent molecules by connecting smaller fragments known to bind distinct regions of a target site. We formulate this as a molecular inpainting task: given a known ligand, we fix the identity and coordinates of two fragments and sample its missing parts consistent with both geometry and reaction grammar.

As case studies, we select several FDA-approved, hard-to-synthesize small molecules with experimental crystal structures bound to different target proteins: human plasma kallikrein (PDB: 7N7X), multidrug-resistant HIV protease 1 (PDB: 4EYR), and human cyclin-dependent kinase 6 (PDB: 5L2S). Each ligand contains at least two of our building blocks. At sampling time, we condition on the substructure match by keeping fixed fragments denoised and interpolating the remaining coordinates (Appendix B.18).

Generated molecules are evaluated with AutoDock Vina (Figure 4) (Eberhardt et al., 2021). SYNCO-GEN consistently produces molecules with docking scores on par with or better than the native ligand while satisfying constraints on the presence of specific building blocks. AlphaFold3 (Abramson et al., 2024b) predictions on selected protein-ligand pairs show similar binding positions in the selected pockets as well (Figures 4 and 17). Crucially, unlike existing approaches (Schneuing et al., 2024; Igashov et al., 2024), the model links fragments using building blocks *and* reactions to ensure streamlined synthetic routes of the designs (Table 8 and Figure 18).

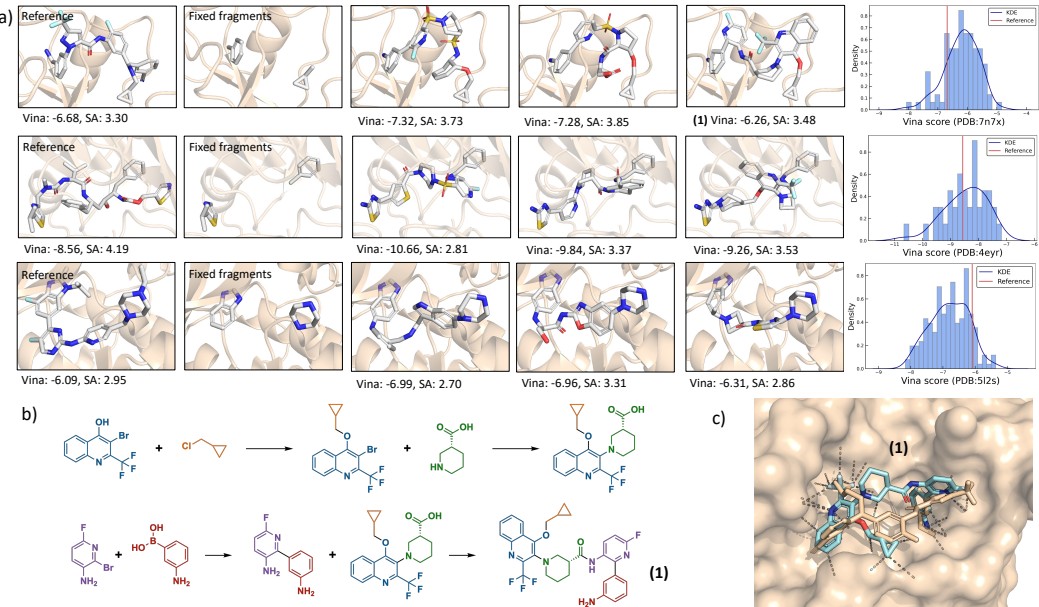

Figure 4: **Molecular inpainting.** a) Fragment linking with three ligands in the PDB that contain substructure matches with our building blocks. For each structure, we show three examples of linkers generated by SYNCOGEN and the distribution of Vina docking scores (lower is better). b) Proposed synthesis pathway for molecule **(1)** sampled from our model and c) structure of **(1)** (blue) docked onto PDB 7N7X using AlphaFold3 compared against the PDB ligand (beige).

Using SYNCOGEN for fragment-linking does not require retraining, although validities and energies can be improved with motif-scaffolding fine-tuning (Table 8). We benchmarked SYNCOGEN against the state-of-the-art, purpose-built fragment-linking model DiffLinker (Igashov et al., 2024). SYNCOGEN is the only method that produces synthesizable molecules with 58-79% retrosynthesis solve rate (0% for DiffLinker, Table 8). Compared to DiffLinker, our molecules have lower interaction energies, no disconnected fragments, reduced hard-to-synthesize features (Table 9), and similar PoseBuster validity rate. The synthesizable inpainted molecules now enables wet-lab tasks such as scaffold hopping, analog generation, or PROTAC design (Békés et al., 2022; Chirnomas et al., 2023).

## 5.3 AMORTIZED PHARMACOPHORE CONDITIONING

We evaluate SYNCOGEN on amortized design of *de novo* small-molecule binders conditioned solely on pharmacophore profiles (Sections 3.3 and 4.1). This setting avoids any external reward models (which can encourage reward hacking) and instead asks the generator to directly realize 3D arrangements of *interaction features* that are compatible with a target pocket or reference ligand. Pharmacophore types and positions are visible to the model during training. To aid generalization, we randomly sample a maximum of 7 pharmacophore features during data loading.

We evaluate on three hard-to-synthesize reference ligands with disease-relevant targets: ozanimod, scopolamine, and TR-107 (PDB: 7EW0, 8CVD, 7UVU), and seven targets from the LIT-PCBA benchmark (Tran-Nguyen et al., 2020): 2IOK, 2P15, 2V3D, 3ZME, 4ZZN, 5FV7, and 5L2M. We compare SYNCOGEN against three baselines. ShEPhERD (Adams et al., 2025) is a state-of-the-art 3D generator conditioned on pharmacophore interaction profiles but does not enforce synthesizability. SynFormer (Gao et al., 2024) generates synthesizable *2D* molecules; we condition it on native ligands for analogue generation. CGFlow (Shen et al., 2025) generates synthesis pathways with 3D poses. For CGFlow, we use the pocket-conditioned reward from Shen et al. (2024) to align with our amortized sampling setup (CGFlow-ZS). For each target, we generate 100 molecules per method based on the cognate ligand pharmacophore profile and dock valid samples with Autodock Vina.

On average, SYNCOGEN produces *de novo* molecules with better or competitive docking scores compared to ShEPhERD, CGFlow-ZS, SynFormer, and the native ligand (Figure 5). Our top samples surpass all baselines in *8 out of 10 targets*. Qualitatively, SYNCOGEN molecules dock to the same pocket and replicate key pharmacophoric contacts of the known ligand with a high degree of shape overlap (Figure 20). Compared to the 3D method ShEPhERD, SYNCOGEN -generated molecules

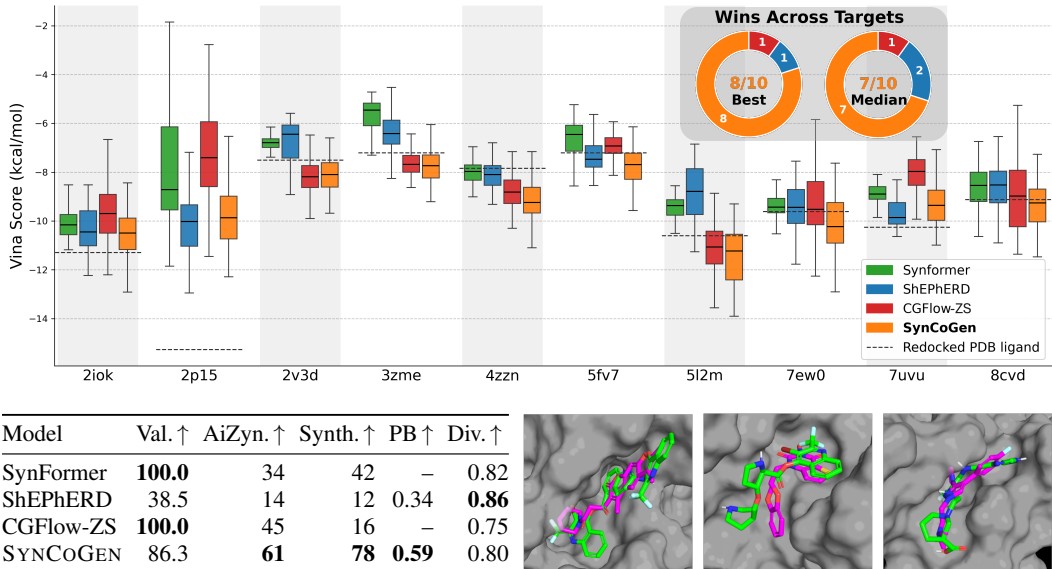

Figure 5: **Pharmacophore-conditioned generation**. *Top*: Docking score comparison on 10 targets from the PDB/LIT-PCBA benchmark (lower is better). Inset: target wins by method, where SYNCO-GEN achieves the best docking score on 8/10 targets (best sample) and 7/10 (median). *Bottom left*: Aggregated conditional generation metrics for all 10 targets. *Bottom right*: Docked SYNCOGEN-generated molecules (green) overlaid with PDB ligand (magenta) for 5L2M, 5FV7 and 3ZME.

| Model | Val. ↑ | AiZyn. ↑ | Synth. ↑ | PB ↑ | Div. ↑ |
|---|---|---|---|---|---|
| SynFormer | **100.0** | 34 | 42 | – | 0.82 |
| ShEPhERD | 38.5 | 14 | 12 | 0.34 | **0.86** |
| CGFlow-ZS | **100.0** | 45 | 16 | – | 0.75 |
| SYNCOGEN | 86.3 | **61** | **78** | **0.59** | 0.80 |

have markedly higher RDKit validity and PoseBusters validity rate (by 45% and 25%, respectively), indicating more chemically and geometrically plausible structures. Most importantly, across all baselines, including synthesis-constrained ones, SYNCOGEN achieves significantly better retrosynthesis solve rates (by 15-65%) and reduces hard-to-synthesize features (Table 10), while maintaining comparable diversity. These results suggest that the added complexity of generating synthesizable molecules *and* their 3D poses within our synthesis-constrained search space (see Figure 19) offers an *amortized* way to design high-affinity, readily synthesizable molecules *de novo*.

## 6 CONCLUSION

In this work, we introduced SYNCOGEN, a multimodal generative model that jointly samples building-block reaction graphs and atomic coordinates. Our chemistry-aware training procedures enable this model to learn to design synthesizable molecules directly in Cartesian space. To support this framework, we curated SYNSPACE, a new dataset family comprising 1.2M readily synthesizable molecules paired with 7.5M low-energy 3D conformations.

SYNCOGEN achieves state-of-the-art performance across 3D molecular generation benchmarks, while natively returning a tractable synthetic route for each structure. Crucially, SYNCOGEN establishes a new standard for *zero-shot* target-conditional design: without target-specific retraining or external rewards, the model generates strong predicted binders in 3D *and* ensures synthesizable chemistry - demonstrated here through both fragment-linking and pharmacophore-conditioned generation.

The design space of SYNCOGEN is not limited to SYNSPACE. Our code base supports custom building blocks and reactions and finetuning/retraining of our models. Looking forward, future works should experimentally validate the synthesis and binding of these de novo designs to establish the practical impact of 3D-conditioned synthesizable molecular generation for drug discovery.

### ACKNOWLEDGMENTS

The authors thank Joey Bose (University of Oxford), Stephen Lu (McGill University), and Francesca-Zhoufan Li (Caltech) for helpful discussions. This work is enabled by high-performance computing at the Digital Research Alliance of Canada and Mila.

## ETHICS STATEMENT

While intended for research in drug discovery, any generative chemistry system has dual-use risk (e.g., suggesting toxic or hazardous compounds). We mitigate this by constraining generation to commercially available building blocks and a limited set of high-yield reaction templates, representing products as explicit reaction graphs, which enables expert review of routes.

## REPRODUCIBILITY STATEMENT

We provide code for this study, including end-to-end training and sampling scripts for the joint multi-modal model, configuration files, evaluation pipelines that reproduce the metrics, and data preparation code to regenerate the conformer sets and pharmacophore features. We also release pretrained checkpoints and commands to reproduce: unconditional generation, fragment-linking inpainting, and pharmacophore-conditioned sampling. Our repository contains simple commands to generate a new training dataset given custom reactions and building blocks.

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

## A  CHEMISTRY AND DATASET DETAILS

### A.1  BUILDING BLOCKS AND REACTIONS

For the small vocabulary, the 93 selected commercial building blocks and their respective reaction centers are shown in Figure 6. For chemical reactions, we focused on cross-coupling reactions to link fragments together. We chose 8 classes of robust reactions, which can be subdivided into 19 types of reaction templates, see Figure 7. The remaining building blocks and reactions that define the large vocabulary are shown in Figure 8 and Figure 9, respectively. We note that our reaction modeling is simplified. For example, boronic acids in building blocks ($B(OH)_2$) are replaced with boranes ($BH_2$); we do not consider the need for chemical protection on certain functional groups (e.g. N-Boc); we do not consider directing group effects or stoichiometry when multiple reaction centers are available; we do not consider macrocycles. These edge cases are limitations of the current method, but they are comparably minimal through the careful curation of building blocks to avoid such infeasible chemical reactions.

### A.2  GRAPH GENERATION

**Helper definitions.**   We annotate each building block with its reaction center atom indices $\mathcal{V}(b) \subseteq V(b)$ and its and each intrinsic atom-level graph by $H(b) := \big(V(b), L(b)\big)$, where $V(b)$ is the set of atoms in $b$ and $L(b) \subseteq V(b) \times V(b)$ is the set of covalent bonds internal to the block. Each reaction template $r$ is annotated with a Boolean tuple $\big((l_A(r), l_B(r)\big) \in \{0, 1\}^2$ describing whether reagent $A$ or reagent $B$ in $r$, respectively, contains a leaving atom.

Given the current atom graph $G_a = (V_a, L_a)$ and an atom $v \in V_a$ of degree 1, the routine UNIQUENEIGHBOR$(v)$ returns the *single* atom $u \in V_a$ such that $(u, v) \in L_a$. Throughout the vocabulary, every leaving-group center has exactly one neighbour.

A reaction template $r$ is considered compatible with $(b_i, v)$ and $(\tilde{b}, \tilde{v})$ if it queries for first and second reagent substructures that match $(b_i, v)$ and $(\tilde{b}, \tilde{v})$, respectively.

Lastly, while the model is compatible with reactions containing more leaving groups, we do not consider them as the dataset construction requires custom atom attribution between reactants and products.

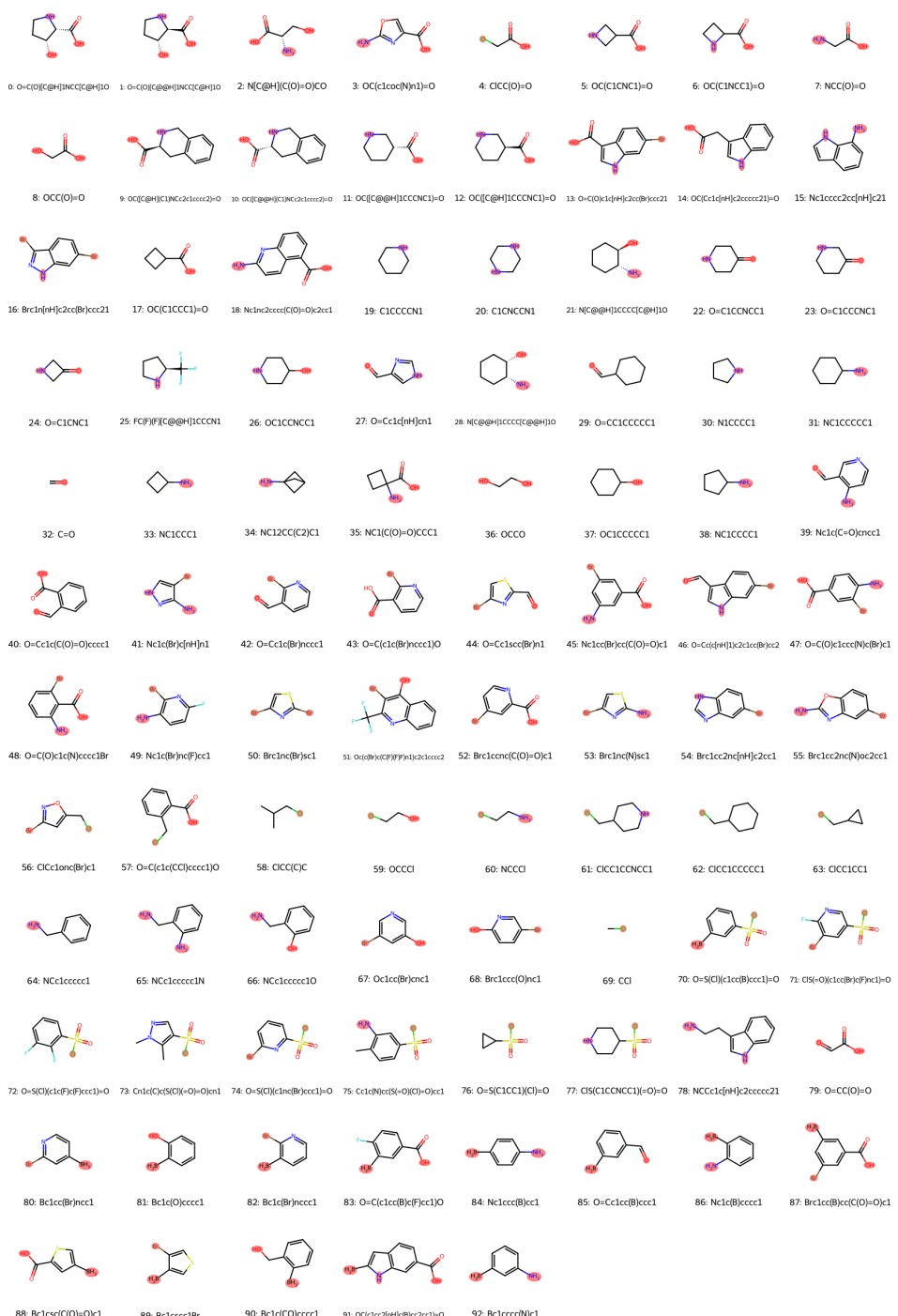

Figure 6: List of building blocks for the small vocabulary, their respective reaction centers (in red), and their canonical SMILES representation.

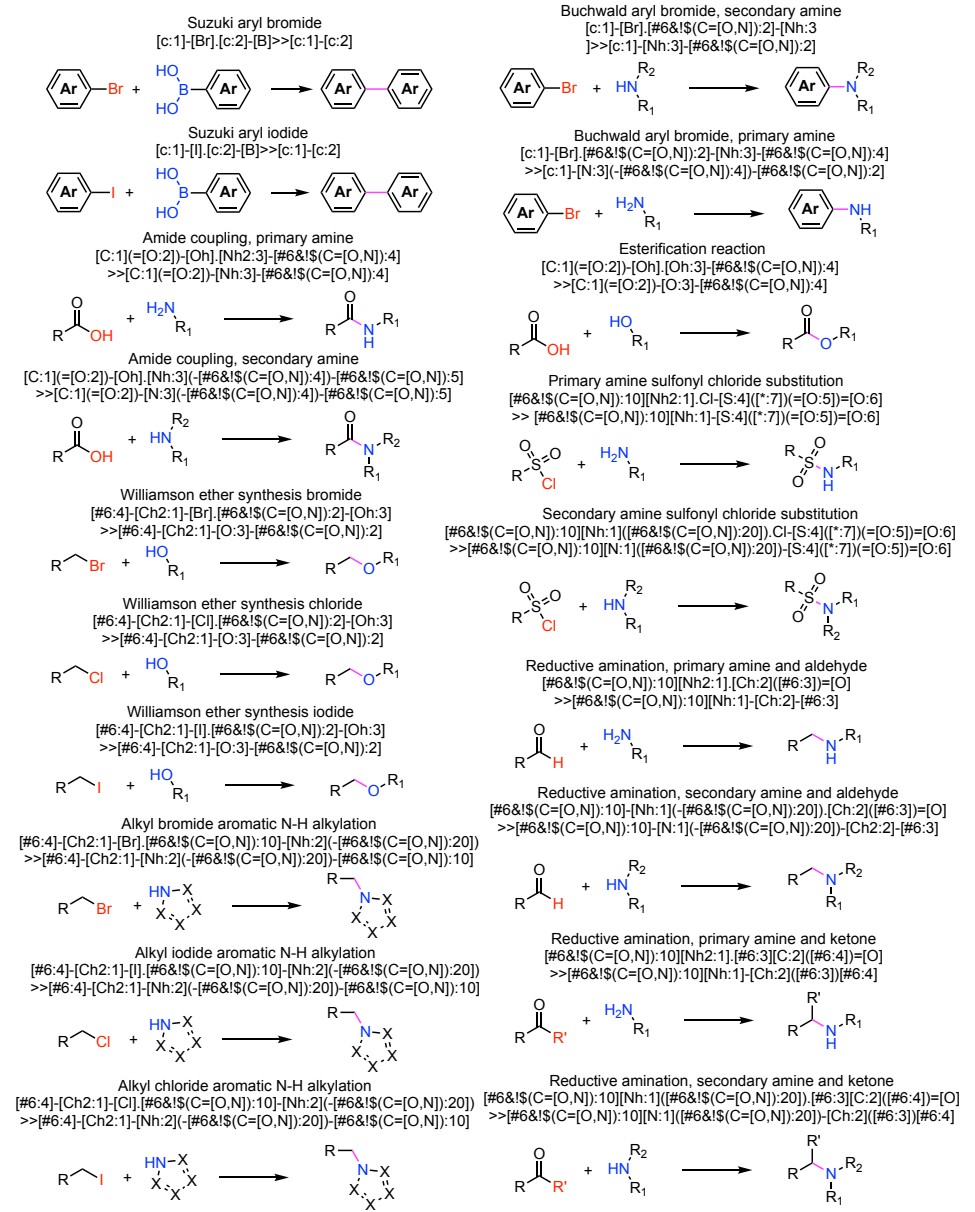

Figure 7: List of chemical reactions for the small vocabulary used to connect building blocks and their SMARTS representation. Newly formed bonds are highlighted in pink.

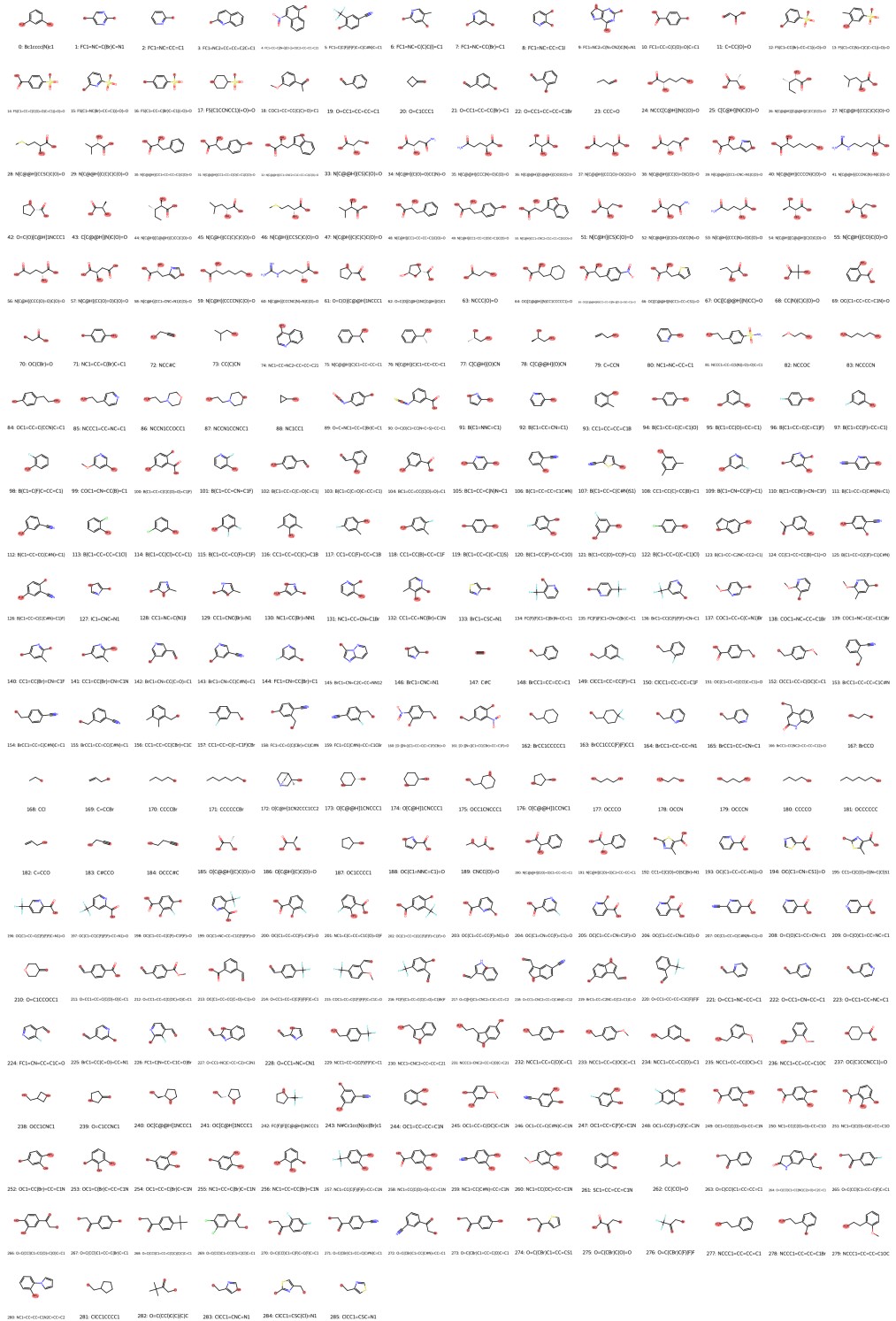

Figure 8: The additional building blocks for the large vocabulary, their respective reaction centers (in red), and their canonical SMILES representation. The large vocabulary also includes all building blocks from the small vocabulary.

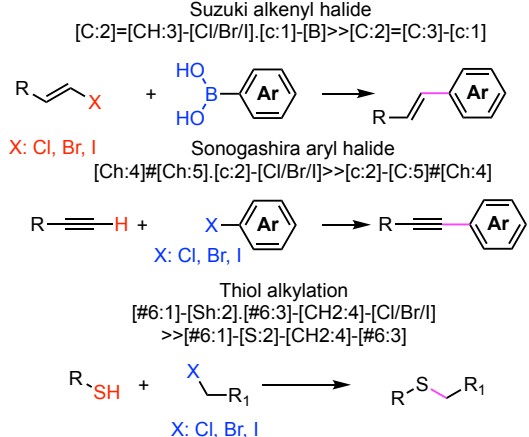

Figure 9: The additional of chemical reactions (from 3 classes) for the large vocabulary used to connect building blocks and their SMARTS representation. Newly formed bonds are highlighted in pink. The large vocabulary also includes all reactions from the small vocabulary.

---

**Algorithm 1** Fragment-by-fragment assembly with COUPLE

---

**Inputs:** vocab $\mathcal{B}$, reactions $\mathcal{R}$, depth limit $T$
**Output:** atom graph $G_a$, building block graph $G_f = (X, E)$

1: **function** COUPLE$(G_a,\ b_i,\ \tilde{b},\ r,\ (v_i, \tilde{v}))$
2:      append all atoms and bonds of $H(\tilde{b})$ to $G_a$      ▷ **1. Handle leaving groups**
3:      **if** $l_A(r) = 1$ **then**      ▷ $v_i$ leaves in reagent A
4:          $u_i \leftarrow$ UNIQUENEIGHBOR$(v_i)$
5:          delete atom $v_i$ (and its bond) from $G_a$
6:          $v_i \leftarrow u_i$      ▷ reroute to neighbour
7:      **end if**
8:      **if** $l_B(r) = 1$ **then**      ▷ $\tilde{v}$ leaves in reagent B
9:          $u_t \leftarrow$ UNIQUENEIGHBOR$(\tilde{v})$
10:         delete atom $\tilde{v}$ (and its bond) from $G_a$
11:         $\tilde{v} \leftarrow u_t$      ▷ reroute to neighbour
12:      **end if**
13:      add covalent bond between $v_i$ and $\tilde{v}$      ▷ **2. Add the cross-bond**
14:      **return** $G_a$
15: **end function**
16: $b_0 \leftarrow$ UniformPick$(\mathcal{B})$;    $G_a \leftarrow H(b_0)$;    $G_f \leftarrow (b_0)$
17: **for** $t = 1$ **to** $T$ **do**
18:      $L \leftarrow$ enumerate compatible 5-tuples $\langle b_i, v, r, \tilde{b}, \tilde{v} \rangle$
19:      **if** $L = \varnothing$ **then break**
20:      **end if**
21:      $(b_i, v, r, \tilde{b}, \tilde{v}) \leftarrow$ UniformPick$(L)$
22:      $e \leftarrow (r, v, \tilde{v})$
23:      $G_a \leftarrow$ COUPLE$(G_a, b_i, \tilde{b}, r, (v, \tilde{v}))$
24:      $G_f \leftarrow G_f \cup \left(b_i \xrightarrow{e} \tilde{b}\right)$
25: **end for**
26: **return** $(G_a, G_f)$

---

## A.3 SYNSPACE STATISTICS

Table 2: Average molecular properties of SYNSPACE and SYNSPACE-L datasets, in comparison with GEOM-Drugs (Axelrod & Gomez-Bombarelli, 2022)

| Property | SYNCOGEN | SYNSPACE-L | GEOM Drugs |
|---|---|---|---|
| Molecular Weight | 492.16 | 476.40 | 355.83 |
| Number of Heavy Atoms | 33.74 | 32.99 | 24.86 |
| Octanol–Water Partition Coefficient (Log P) | 2.44 | 3.01 | 2.91 |
| Number of Hydrogen Bond Donors | 2.75 | 3.30 | 1.19 |
| Number of Hydrogen Bond Acceptors | 6.74 | 6.25 | 4.83 |
| Quantitative Estimate of Drug-likeness | 0.43 | 0.36 | 0.65 |
| Fraction of $sp^3$ Carbons | 0.41 | 0.37 | 0.30 |
| Topological Polar Surface Area | 111.32 | 110.08 | 73.73 |
| Number of Rotatable Bonds | 6.95 | 8.92 | 4.90 |
| SAScore | 3.34 | 3.28 | 2.51 |
| **Murcko Scaffold Number** | **443458** | **333180** | **92955** |

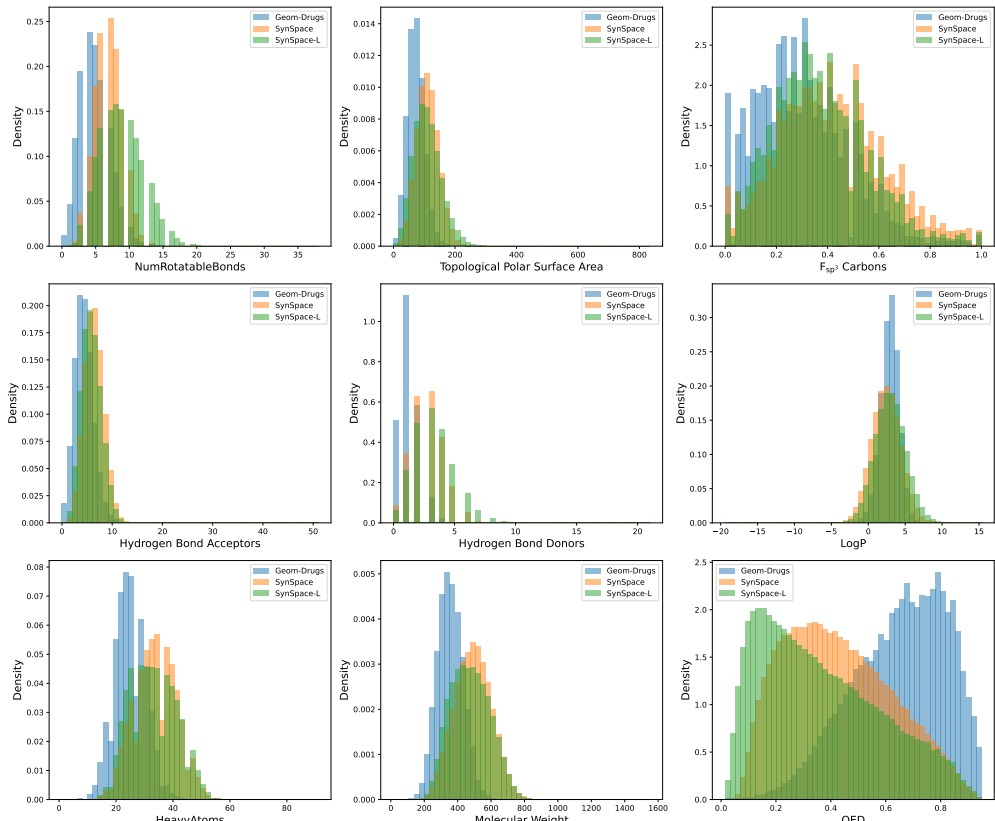

Figure 10: Distribution of SYNSPACE and SYNSPACE-L molecular property statistics, as compared to GEOM Drugs.

# B  METHOD DETAILS

## B.1  SIMPLIFIED TRAINING WORKFLOW

Below we provide a simplified illustration of the SYNCOGEN training process. For visual clarity, we describe the procedure for a single building block.

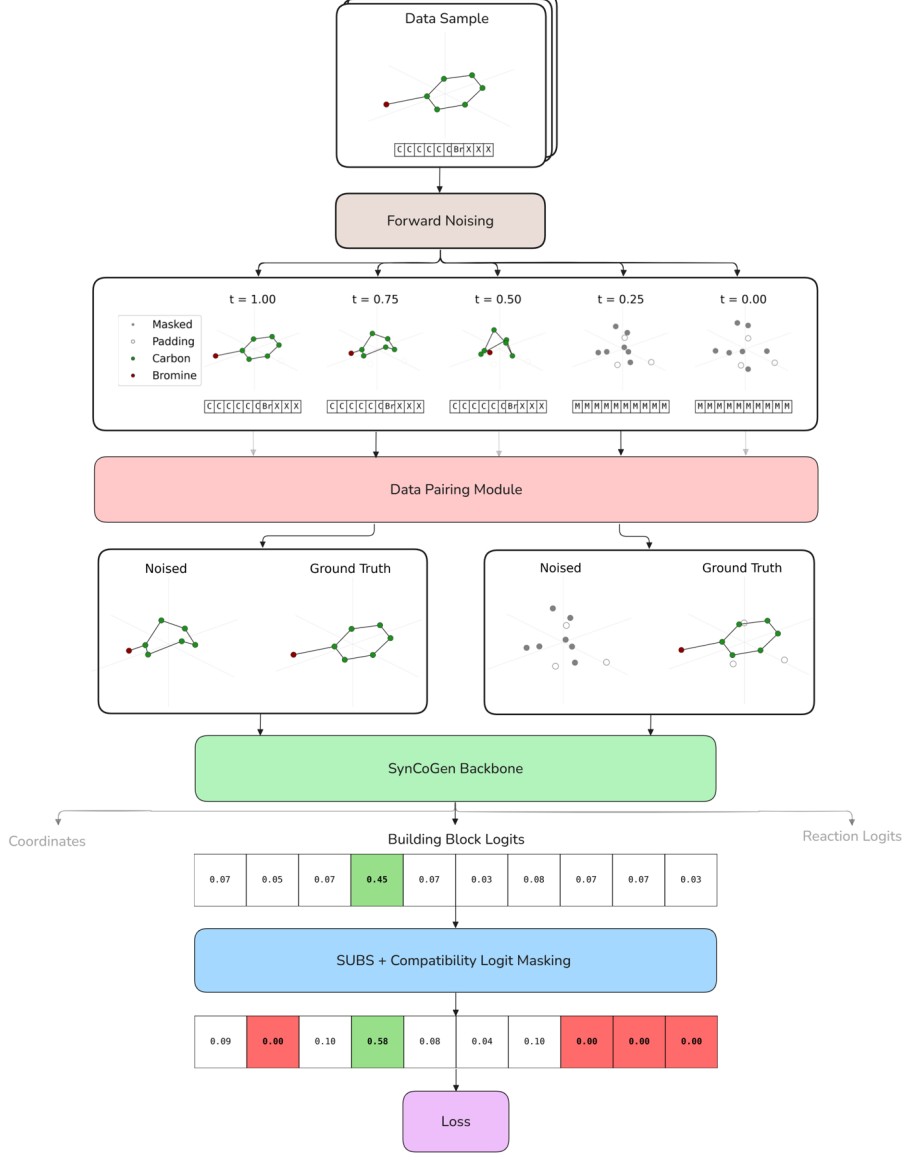

Figure 11: Simplified training workflow for SYNCOGEN using a single bromobenzene as an example.

Data are passed through the model during training according to the following process:

1. **Noise injection.** Coordinate positions and building block identities are noised/masked. If a building block becomes masked, padding atoms are added to its coordinates to match the maximum number of atoms in any building block within the vocabulary $M$. In this example, $M = 10$.

2. **Ground-truth preparation.** To keep the number of atoms consistent, the data pairing module (Appendix B.6) generates ground-truths with or without padding atoms and re-centers them accordingly. Note that padding atom positions are identical in the noisy

coordinates (sampled from the Gaussian prior) and the ground truth. Here, we encourage the model to disregard atoms that are unlikely to assemble into the true molecule when the building block is unknown.

3. **Backbone processing.** The noised building blocks are passed through the backbone, which outputs building block logits, reaction logits and coordinates; we exemplify this for building blocks in the diagram above. The correct index is highlighted in green.

4. **Index masking.** The logits are processed by the SUBS parameterization module introduced by Sahoo et al. (2024) and the compatibility logit masking module described in Appendix B.2 to eliminate probability mass allocated to incompatible or impossible indices.

5. **Loss computation.** Negative log-likelihoods are computed over the modified logits.

**Remark.** Steps 1 and 2, in particular, ensure that model inputs containing masked building blocks during training remain consistent with the information available for a corresponding example at sampling time. When a building block is not known, neither is the number of atoms it contains. The absence of padding atoms during training would require direct selection of atom counts per building block at sampling time, which constitutes a strong constraint on the building block identities and severely limits design flexibility.

## B.2 COMPATIBILITY LOGIT MASKING

Below we provide a simplified illustration of the SYNCOGEN compatibility masking procedure for building blocks. For visual clarity, selection of building block attachment points is implicit and reactions are denoted by a single one-hot item $r$, rather than a triple $(r, v_i, v_j)$.

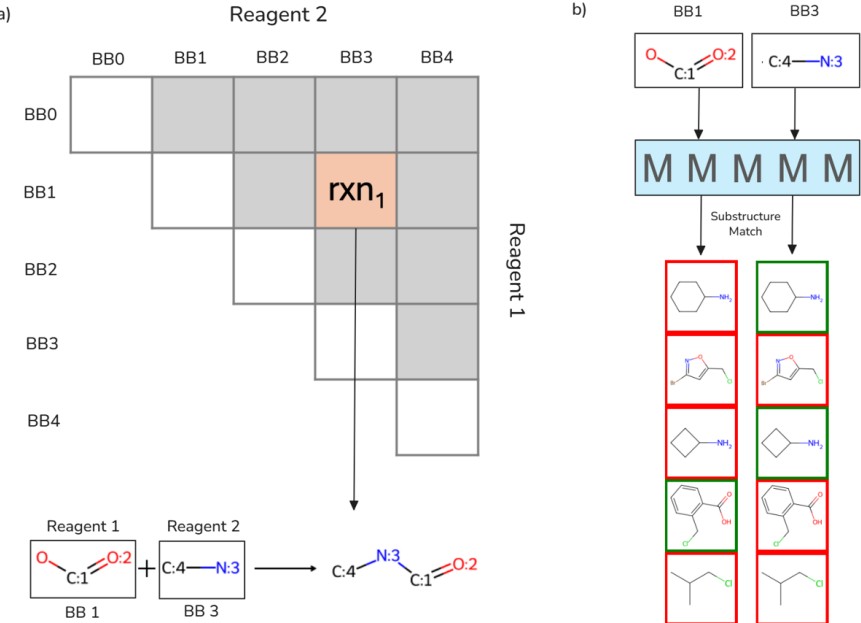

Figure 12: **Compatibility masking regime for building blocks.** Gray and white squares indicate "masked" and "no edge", respectively. a) A denoised item at position $(1, 3)$ in $E$ denotes that a reaction $r_1$ has been selected between building block 1 and 3 in $X$. b) In $X$, the vocabulary is queried for substructure matches to reagents 1 and 2 of $r_1$ at building block indices 1 and 3 respectively, and logits corresponding to incompatible building blocks are set to 0.

## B.3 SAMPLING EDGE LOGIT MASKING

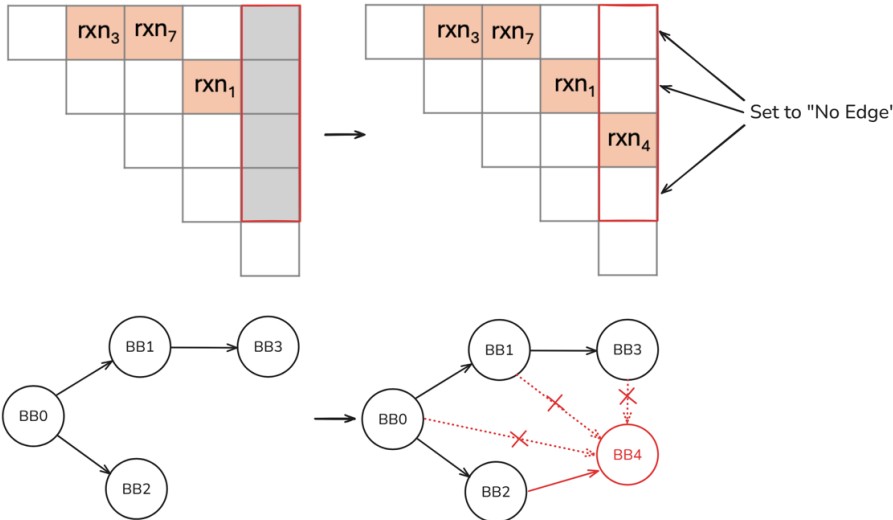

Figure 13: **Sampling constraints for edge denoising.** When a reaction is denoised at position $(2, 4)$ in $E$, all other incoming edges to building block index 4 are set to "no-edge"—this is a valid assumption as SynSpace does not contain macrocycles.

## B.4 BUILDING BLOCK-LEVEL REPRESENTATIONS

Let $X \in \{0,1\}^{N \times |\mathcal{B}|+1}$ be a one-hot matrix where the $i^{\text{th}}$ row encodes the identity of the $i^{\text{th}}$ building block, and let $E \in \{0,1\}^{N \times N \times |\mathcal{R}| V_{max}^2 + 2}$, where $V_{max} = \max_b |\mathcal{V}(b)|$. A non-zero entry $E_{ijr(v_i,v_j)} = 1$ signals that block $i$ (center $v_i$) couples to block $j$ (center $v_j$) via reaction $r$. Graphs $(X, E)$ belonging to molecules containing $n < N$ building blocks are padded to $N$.

**Reserved Channels.** We reserve a dedicated *masked* (absorbing) token in both vocabularies:

$$\pi_X \in \{0,1\}^{|\mathcal{B}|}, \qquad \pi_E \in \{0,1\}^{|\mathcal{R}| V_{\max}^2}, \tag{5}$$

where $\pi_X$ (resp. $\pi_E$) is the one-hot vector whose single 1-entry corresponds to the masked node (resp. edge) channel. Besides the masked channel, we keep a dedicated *no-edge* channel, encoded by the one-hot vector

$$\lambda_E \in \{0,1\}^{|\mathcal{R}| V_{\max}^2}, \tag{6}$$

so every edge slot may take one of three mutually exclusive states: a concrete coupling label, the no-edge token $\lambda_E$, or the masked token $\pi_E$.

## B.5 ATOM-LEVEL REPRESENTATIONS

The SEMLAFLOW (Irwin et al., 2025) architecture propagates and updates invariant and equivariant features at the atom level. To ensure consistency with this framework, we calculate for each input graph $(X_t, E_t)$ atom-level one-hot atom and bond features. Crucially, these features must be flexible to arbitrary masking present in $X_t$ and $E_t$. With this in mind we set each atom feature $X_t^{atom}[i, a]$ to a concatenation of one-hot encodings

$$X_t^{atom}[i, a] = \Big( \underbrace{\delta_{\text{sym}(i,a)}}_{\text{9-way one-hot}}, \ \mathbb{1}[\text{ring}(i, a)], \ \mathbb{1}[a \in \mathcal{V}(X_i)] \Big) \in \{0, 1\}^{9+2}, \tag{7}$$

where $\delta_{\text{sym}}(i, a)$ is the one-hot vector over possible atom types (C, N, O, B, F, Cl, Br, S, [MASK]) and ring$(i, a)$ denotes whether or not the atom is a member of a ring. Similarly, we calculate a bond

feature matrix

$$E_t^{\text{atom}}[a_i, a_j] = \begin{cases} \delta_{\text{order}}(a_i, a_j), & \text{bond is present,} \\ \mathbf{0}_5, & \text{otherwise.} \end{cases} \tag{8}$$

where $\delta_{\text{order}}(a_i, a_j)$ is the one-hot tensor over possible bond orders (single, double, triple, aromatic, [MASK]) between $a_i$ and $a_j$. $E_t^{atom}$ is populated by loading the known bonds and respective bond orders within denoised building blocks. If some building block $X_i$ is noised, all edges between its constituent atoms $E_t^{atom}[i : i + M, i : i + M]$ are set to the masked one-hot index. For graphs $(X_t, E_t)$ corresponding to valid molecules in which all nodes and edges are denoised, we simply obtain the full bond feature matrix from the molecule described by $(X_t, E_t)$.

## B.6 DATA PAIRING

---

**Algorithm 2** PAIRDATA$\big(C_0, S_0, C_1, t, X_t\big)$

---

**Input:** $C_0$ (clean coordinates), $S_0$ (atom mask), $C_1$ (prior sample), $t \in [0, 1]$, $X_t$ (partially masked nodes)
**Output:** $\tilde{C}_0$ (re-centered ground truth), $C_t$ (interpolated noisy coords)

1:  $\mathcal{D}_t \leftarrow \{ i \mid X_t[i] \neq \pi_X \}$                                          ▷ denoised blocks
2:  $S_t[i, a] \leftarrow \mathbf{1}[i \notin \mathcal{D}_t \vee a \in \mathcal{A}_i]$                          ▷ visibility
3:  $\tilde{C}_1 \leftarrow C_1 - \bar{C}_{1_{S_t}}$
4:  $\tilde{C}_0 \leftarrow \text{ZEROTENSOR}()$
5:  **for all** $(i, a)$ **do**
6:     **if** $S_0[i, a] = 1$ **then**
7:        $\tilde{C}_0[i, a] \leftarrow C_0[i, a] - \bar{C}_{1_{S_t}}$
8:     **else if** $S_t[i, a] = 1$ **then**                              ▷ dummy atom
9:        $\tilde{C}_0[i, a] \leftarrow \tilde{C}_1[i, a]$
10:    **end if**
11: **end for**
12: $C_t \leftarrow (1 - t)\,\tilde{C}_0 + t\,\tilde{C}_1$
13: **return** $\big(\tilde{C}_0, C_t\big)$

---

Here, $\mathcal{A}_i$ is the set of all atom indices $a$ that constitute true atoms in $X_0$. Note that $S_t = S_0$ for all $t$ where $X_t$ contains no masked building blocks.

**Note: Non-Equivariance.** Our data pairings result in both $C_0$ and $C_t$ that are properly centered according to atoms that are possibly valid at time $t$. It is important to note that under this scheme, while the model is $SE(3)$-equivariant with respect to the system defined by the partial mask $S_t$, it is not equivariant with respect to the orientation of the molecule itself unless $\mathcal{D}_t^c = \varnothing$, as the presence and temporary validity of masked dummy atoms offsets the true atom centering and thus breaks both translational and rotational equivariance.

## B.7 TRAINING ALGORITHM

---

**Algorithm 3** Training step for SYNCOGEN

---

1:  $t \sim \mathcal{U}(0, 1)$
2:  $(X_t, E_t) \leftarrow q_t(X_0, E_0)$
3:  $C_1 \sim \mathcal{N}(0, I)$
4:  $(\tilde{C}_0, \tilde{C}_t) \leftarrow \text{PAIR}(C_0, S_0, C_1, t, X_t)$        ▷ center and interpolate coordinates (Algorithm 2)
5:  $(L_t^X, L_t^E, \hat{\tilde{C}}_0^t) \leftarrow f_\theta(X_t, E_t, \tilde{C}_t, n, t)$
6:  $\mathcal{L} \leftarrow \mathcal{L}_{\text{graph}} + \mathcal{L}_{\text{MSE}} + \mathcal{L}_{\text{pair}}$                  ▷ total loss (Appendix B.16)
7:  $\theta \leftarrow \theta - \eta - -bla_\theta \mathcal{L}$

---

## B.8 SAMPLING ALGORITHM

---

**Algorithm 4** Sampling procedure for SYNCOGEN

---

1: $n \sim \text{Cat}(\pi_{\text{frag}})$; $(X_1, E_1) \leftarrow (\pi_X, \pi_E)$; $S_1[i, a] \leftarrow \mathbf{1}[i < n]$       ▷ draw $n$, initialize masks
2: $C_1 \sim \mathcal{N}(0, I)$; $\tilde{C}_1 \leftarrow C_1 - \bar{C}_{1, S_1}$       ▷ center Gaussian prior by initial mask
3: **for** $t = 1$ **down to** $0$ **do**
4:      $\tilde{C}_t \leftarrow C_t - \bar{C}_{t, S_t}$;
5:      $(L_t^X, L_t^E, \hat{\tilde{C}}_0^t) \leftarrow f_\theta(X_t, E_t, \tilde{C}_t, n, t)$
6:      $\tilde{L}_t^E \leftarrow \text{SAMPLEEDGES}(L_t^E, n)$       ▷ enforce one parent per building block (Algorithm 5)
7:      $X_{t-\Delta t} \leftarrow \text{CATSAMPLE}(L_t^X)$; $E_{t-\Delta t} \leftarrow \text{CATSAMPLE}(\tilde{L}_t^E)$       ▷ take reverse step (Appendix B.11)
8:      $C_{t-\Delta t} \leftarrow C_t + \Delta t(\hat{\tilde{C}}_0^t - \tilde{C}_t)$
9:      $(X_t, E_t, C_t, S_t) \leftarrow (X_{t-\Delta t}, E_{t-\Delta t}, C_{t-\Delta t}, S_{t-\Delta t})$
10: **end for**
11: $(L^X, L^E, \hat{\tilde{C}}_0) \leftarrow f_\theta(X_0, E_0, \tilde{C}_0, n, 0)$       ▷ final deterministic denoise ($t = 0$)
12: $\hat{X}_0 \leftarrow \arg\max_k L_\theta^X[\cdots, k]$; $\hat{E}_0 \leftarrow \arg\max_k L_\theta^E[\cdots, k]$; $\hat{C}_0 \leftarrow \hat{\tilde{C}}_0 - \bar{\hat{\tilde{C}}}_{0, S_0}$
13: **return** $(\hat{X}_0, \hat{E}_0, \hat{C}_0)$

---

## B.9 INFERENCE-TIME EDGE CONSTRAINTS

Let $E_\theta^t \in [0, 1]^{n \times n \times |\mathcal{R}| V_{\max}^2}$ be the soft-max edge probabilities produced at step $t$. The routine below resolves the unique parent for every building block column $j > 0$ and returns a probability tensor $\tilde{E}_\theta^t$ with exactly one non–zero entry per column.

---

**Algorithm 5** SAMPLEEDGES$\left(E_\theta^t, n\right)$

---

**Input:** edge probabilities $E_\theta^t$
**Output:** pruned probabilities $\tilde{E}_\theta^t$

1: $\tilde{E}_\theta^t \leftarrow \mathbf{0}$
2: **for** $j = 1$ **to** $n - 1$ **do**
3:      $(i_j, e_j) \sim \text{Cat}\left(\{E_\theta^t[i, j, e] \mid 0 \le i < j\}\right)$
4:      $\tilde{E}_\theta^t[i_j, j, e_j] \leftarrow 1$
5: **end for**
6: **return** $\tilde{E}_\theta^t$

---

$\tilde{E}_\theta^t$ is then symmetrized and fed to the discrete reverse sampler described in Appendix B.11.

## B.10 BUILDING BLOCK LOGIT PREDICTIONS

The SEMLAFLOW(Irwin et al., 2025) backbone outputs atom–atom edge features $E_\theta^{\text{atom}} \in \mathbb{R}^{B \times (NM) \times (NM) \times d_{\text{edge}}}$. To obtain building block-level tensors, we apply two parallel 2-D convolutions (one for nodes, one for edges) with stride $M$, followed by MLP classifiers that map the pooled features back to their original one-hot vocabularies. Note that the presented model is trained to predict a maximum of 5 building blocks, where the sizes of the molecules (average 566 Da) are near upper limits of molecular weights for typical drug like molecules.

**Stride-pooled convolution.** Let $d_{edge}$ be the latent edge feature dimension. Each stream uses the block

$$\text{Conv2d}(d_{\text{edge}} \to d_{\text{edge}}, \ k = M, \ s = M) \xrightarrow{\text{SiLU}} \text{Conv2d}(d_{\text{edge}} \to d_{\text{edge}}, \ k = 1, \ s = 1), \quad (9)$$

so every $M \times M$ atom patch collapses to a single building block entry. This produces

$$X_{\text{pool}} \in \mathbb{R}^{B \times d_{\text{edge}} \times N}, \qquad E_{\text{pool}} \in \mathbb{R}^{B \times d_{\text{edge}} \times N \times N}. \quad (10)$$

**Node head.** We flatten $X_{\text{pool}}$ along its channel axis, concatenate the residual building block one-hot matrix $X_t$, and pass the result through a two-layer MLP to obtain

$$L_\theta^{X_t} \in \mathbb{R}^{B \times N \times |\mathcal{B}|}. \tag{11}$$

**Edge head.** We concatenate $E_{\text{pool}}$ with the residual building block-edge one-hot tensor $E_t$, apply an analogous two-layer MLP, and symmetrize to produce

$$L_\theta^{E_t} \in \mathbb{R}^{B \times N \times N \times |\mathcal{R}|V_{\max}^2}. \tag{12}$$

**Atom Features.** The SEMLAFLOW(Irwin et al., 2025) backbone additionally outputs atom-level node features $X_\theta^{\text{atom}} \in \mathbb{R}^{B \times (NM) \times d_{\text{node}}}$, which are incorporated into $E_\theta^{\text{atom}}$ via a bond refinement message-passing layer. We find that extracting both building block and edge logits directly from the refined features $E_\theta^{\text{atom}}$ marginally improves performance relative to separately predicting $L_\theta^{X_t}$ from $X_\theta^{\text{atom}}$ and $L_\theta^{E_t}$ from $E_\theta^{\text{atom}}$.

## B.11 DISCRETE NOISING SCHEME

Following (Sahoo et al., 2024), we adopt an absorbing (masked) state noising scheme for $X_0$ and $E_0$:

$$q(X_t \mid X_0) = \text{Cat}\big(X_t; \alpha_t X_0 + (1 - \alpha_t)\pi_X\big), \qquad q(E_t \mid E_0) = \text{Cat}\big(E_t; \alpha_t E_0 + (1 - \alpha_t)\pi_E\big). \tag{13}$$

where $(\alpha_t)_{t \in [0,1]}$ is the monotonically decreasing noise schedule introduced in Section 2.

**Reverse categorical posterior.** For node identities, we have

$$q(X_s \mid X_t, X_0) = \begin{cases} \text{Cat}(X_s; X_t), & X_t \neq \pi_X, \\ \text{Cat}\big(X_s; \dfrac{(1 - \alpha_s)\pi_X + \alpha_s X_\theta^t}{1 - \alpha_t}\big), & X_t = \pi_X, \end{cases} \tag{14}$$

and, analogously, for edge labels

$$q(E_s \mid E_t, E_0) = \begin{cases} \text{Cat}(E_s; E_t), & E_t \neq \pi_E \\ \text{Cat}\big(E_s; \dfrac{(1 - \alpha_s)\pi_E + \alpha_s E_\theta^t}{1 - \alpha_t}\big), & E_t = \pi_E, \end{cases} \tag{15}$$

where $s < t$. Equations (14) and (15) are the direct translation of the reverse denoising process described by (Sahoo et al., 2024) into SYNCOGEN's node–edge representation.

## B.12 NOISE SCHEDULE PARAMETERIZATION

Following MDLM (Sahoo et al., 2024), we parameterize the discrete noising schedule via $\alpha_t = e^{-\sigma(t)}$, where $\sigma(t) : [0, 1] \to \mathbb{R}^+$. In all experiments, we adopt the **linear schedule**:

$$\sigma(t) = \sigma_{\max}t, \tag{16}$$

where $\sigma_{\max}$ is a large constant; we use $\sigma_{\max} = 10^8$ as in the original MDLM setup.

**Edge Symmetrization.** After drawing the upper-triangle entries of the one-hot edge tensor $E_s$ in either the forward or reverse (de)noising process, we enforce symmetry by copying them to the lower triangle:

$$E_{s,jie} = E_{s,ije}, \qquad 0 \le i < j < n, \ e \in \mathcal{R}V_{\max}^2.$$

## B.13 POSITIONAL EMBEDDINGS

Though SEMLAFLOW(Irwin et al., 2025) is permutationally invariant by design with respect to atom positions, SYNCOGEN dataset molecules require that atom order be fixed and grouped by building block for reconstruction purposes. To enforce this during training, we intentionally break permutation invariance by generating and concatenating to each input coordinate sinusoidal positional embeddings representing both global atom index and building block index.

## B.14 HYPERPARAMETERS

We train SYNCOGEN for 100 epochs with a batch size of 128 and a global batch size of 512. Note that SEMLAFLOW and Midi are trained for 200 epochs, and EQGAT-diff is trained for up to 800 epochs. All models are trained with a linear noise schedule (see Appendix B.12), with the SUBS

parameterization enabled. During training, a random conformer for each molecule is selected, then centered and randomly rotated to serve as the ground-truth coordinates $C_0$. All atomic coordinates are normalized by a constant $Z_c$ describing the standard deviation across all training examples. For the pairwise distance loss $\mathcal{L}_{pair}$, we set $d$ to 3Å, adjusted for normalization. During training, for each recentered input-prior pair $(\tilde{C}_1, \tilde{C}_0)$ we rotationally align $C_1$ to $C_0$. When training with noise scaling and the bond loss time threshold, we set the noise scaling coefficient to 0.2 and the time threshold to 0.25, above which bond length losses are zeroed. When training with auxiliary losses, we set the weights for the pairwise, sLDDT, and bond length loss components to 0.4, 0.4, and 0.2, respectively.

## B.15 COMPUTATIONAL RESOURCES USED

We train all models on 2 H100-80GB GPUs.

## B.16 TRAINING LOSSES

Here, we define several loss terms that have proved useful for stabilizing training on 3-D geometry. By default, SYNCOGEN is trained with $\mathcal{L}_{\text{MSE}}$ and $L_{\text{pair}}$ as coordinate losses.

For a prediction $\left(L_\theta^{X_t}, L_\theta^{E_t}, \hat{\tilde{C}}_0^t\right) = f_\theta(X_t, E_t, \tilde{C}_t, n, t)$, $X_\theta^t = \text{softmax}(L_\theta^{X_t})$, $E_\theta^t = \text{softmax}(L_\theta^{E_t})$:

**Graph loss.** Let $X_0$ and $E_0$ be the clean node and edge tensors. Following the MDLM implementation (Sahoo et al., 2024), we weigh the negative log-likelihood at step $t$ by

$$w_t = \frac{\Delta\sigma_t}{\exp(\sigma_t) - 1}, \qquad \Delta\sigma_t = \sigma_t - \sigma_{t-1}, \ \ \sigma_0 = 0, \tag{17}$$

where $\sigma_t$ is the discrete noise level. The discrete (categorical) loss is then

$$\mathcal{L}_{\text{graph}} = w_t\left(-\log X_\theta^t[X_0] - \log E_\theta^t[E_0]\right), \tag{18}$$

i.e. the cross-entropy between the one-hot ground truth and the predicted distributions for both nodes and edges.

**MSE loss.** Let $S_0 \in \{0,1\}^{N \times M}$ mask the atoms that exist in the clean structure and $C_t$ be the noisy coordinates. Denote $\mathcal{A}_{S_0} = \{(i, a) : S_0[i, a] = 1\}$.

$$\mathcal{L}_{\text{MSE}} = \frac{1}{|\mathcal{A}_{S_0}|} \sum_{(i,a) \in \mathcal{A}_{S_0}} \left\|\hat{C}_0[i, a] - C_0[i, a]\right\|_2^2, \tag{19}$$

**Pairwise loss.**

$$\mathcal{L}_{\text{pair}} = \sum_{\substack{(i,a)<(j,b) \\ \|C_0[i,a] - C_0[j,b]\|_2 \leq d}} S_0[i, a]\, S_0[j, b]\, \left(\|\hat{C}_0[i, a] - \hat{C}_0[j, b]\|_2 - \|C_0[i, a] - C_0[j, b]\|_2\right)^2, \tag{20}$$

where $d$ is the distance cut-off for pairwise terms. The default total loss value for the model is therefore

$$\mathcal{L}_{\text{SYNCOGEN}} = \mathcal{L}_{\text{graph}} + \mathcal{L}_{\text{MSE}} + \mathcal{L}_{\text{pair}}. \tag{21}$$

**Smooth-LDDT loss (Abramson et al., 2024a).** Let $d_{ij}^0 := \|C_0[i] - C_0[j]\|_2$ and $d_{ij}^{\text{pred}} := \|\hat{C}_0[i] - \hat{C}_0[j]\|_2$ be ground-truth and predicted inter-atomic distances, respectively. For each pair of atoms within a 15 Å cutoff in the reference structure, we compute the per-pair score

$$\text{sLDDT}_{ij} = \frac{1}{4} \sum_{k=1}^4 \sigma\left(\tau_k - |d_{ij}^{\text{pred}} - d_{ij}^0|\right), \quad [\tau_1, \tau_2, \tau_3, \tau_4] = [0.5, 1, 2, 4] \text{ Å},$$

where $\sigma(x) = 1/(1 + e^{-x})$ is the logistic function. The smooth-LDDT loss averages $1 - \text{sLDDT}_{ij}$ over all valid pairs,

$$\mathcal{L}_{\text{sLDDT}} = \frac{\sum_{i<j} \mathbb{1}[d_{ij}^0 < 15]\, S_0[i]\, S_0[j] \left(1 - \text{sLDDT}_{ij}\right)}{\sum_{i<j} \mathbb{1}[d_{ij}^0 < 15]\, S_0[i]\, S_0[j]}. \tag{22}$$

**Bond-length loss.** Given a set of intra-fragment bonds $\text{bonds} = \{(p,q)\}$ extracted from the vocabulary, we penalize deviations in predicted bond lengths:

$$\mathcal{L}_{\text{bond}} = \frac{1}{|\text{bonds}|} \sum_{(p,q)\in\text{bonds}} \left| \|\hat{C}_0[p] - \hat{C}_0[q]\|_2 - \|C_0[p] - C_0[q]\|_2 \right|. \tag{23}$$

**Self-Conditioning.** The modified SEMLAFLOW (Irwin et al., 2025) backbone operates on node and edges features at the atomic level, but outputs unnormalized prediction logits $\hat{X}_0 \in \{0,1\}^{N \times |\mathcal{B}|}$ and $\hat{E}_0 \in \{0,1\}^{N \times N \times |\mathcal{R}| V_{max}^2}$. We therefore implement modified self-conditioning for SYNCOGEN that projects previous step graph predictions $\hat{X}_{0_{cond}}$ and $\hat{E}_{0_{cond}}$ to the shape of $X_t^{atom}$ and $E_t^{atom}$ using an MLP.

### B.17 CONFORMER GENERATION

We randomly assembled 50 molecules with the reaction graph and used the standard conformational search (iMTD-GC) in CREST with GFN-FF to find all reference conformers. For both SYNCOGEN and RDKit ETKDG, we sampled 50 conformers per molecule and computed the coverage and matching scores. We used a relatively strict RMSD threshold of $\tau = 0.75$ Å.

Formally, COV is defined as:

$$\text{COV} = \frac{1}{N} \sum_{i=1}^{N} \mathbb{1}\left[ \min_{1 \le j \le M} \text{RMSD}(m_i, g_j) \le \tau \right], \tag{24}$$

where $\mathbb{1}[\cdot]$ is the indicator function, $m_i$ are the $N$ generated conformers and $g_j$ are the $M$ reference conformers. And MAT is defined as:

$$\text{MAT} = \frac{1}{N} \sum_{i=1}^{N} \min_{1 \le j \le M} \text{RMSD}(m_i, g_j). \tag{25}$$

### B.18 MOLECULAR INPAINTING

For the inpainting experiments in Section 5.2, we keep two fragments $\mathcal{D} = \{\mathcal{D}^{(1)}, \mathcal{D}^{(2)}\}$ and their coordinates fixed and sample the remaining part of the molecule. We follow Appendix B.8 and initialize the graph prior $X_1$ with the one-hot encoding of the desired fragment $i$ at a specified node index in the graph (decided at random or based on the structure of the original molecule, so that it matches its scaffold). For each denoised fragment $\mathcal{D}^{(i)}$, we replace its coordinates at each time $t > 0.03$ during sampling by

$$C_t^{(i)} = (1 - t)\, \tilde{C}_0^{(i)} + t\, \tilde{C}_1^{(i)},$$

where $\tilde{C}_0^{(i)}$ and $\tilde{C}_1^{(i)}$ are the centered ground-truth and prior coordinates of fragment $i$, respectively, and all other fragments are updated as shown in Appendix B.8. For any $t \le 0.03$, which for 100 sampling steps amounts to the last three steps in the path, we follow normal Euler steps as shown in Appendix B.8 to allow a refinement of the fixed coordinates in line with the rest of the predicted ones for the rest of the fragments. We empirically observed that this led to molecules with lower average energies.

## C BASELINE COMPARISONS.

### C.1 UNCONDITIONAL GENERATION.

For all baselines, we sampled 1000 molecules with random seeds on an A100 GPU and reported averaged results over three runs.

**SemlaFlow**    We evaluated SemlaFlow using the sampling script in the official codebase on GitHub[1]. We reported results for a model trained on the GEOM (Axelrod & Gomez-Bombarelli, 2022) dataset (by sampling from the checkpoints provided in the repository) and from a model trained on our dataset (see Table 1). We trained SemlaFlow using the default hyperparameters for 150 epochs on a single conformer per molecule.

**EQGAT-diff, MiDi, JODO, FlowMol**    We evaluated EQGAT-diff, Midi, JODO, using their official implementations provided on GitHub[2]. We modified the example sampling script to save molecules as outputted from the reverse sampling, without any post-processing. For MiDi, we evaluated the uniform model. For FlowMol, both CTMC and Gaussian models were evaluated and reported.

## C.2    Conditional Generation.

In the pharmacophore-conditioned generation setting, we compare SYNCOGEN against SynFormer (Jocys et al., 2024), CGFlow (Shen et al., 2025), and ShEPhERD (Adams et al., 2025). SynFormer was conditioned on the native ligand for synthesizable analogue generation. We used the official implementation on GitHub[3], and changed the following inference settings to allow for higher quality designs compared to the default: `search_width=32`, `exhaustiveness=128`, `time_limit=300`. For ShEPhERD, we use the $p(x_1|x_3, x_4)$ conditional setting from the paper experiments where $x_1$ denotes molecular structure, $x_3$ denotes the reference ligand charge surface, and $x_4$ denotes the reference ligand pharmacophore profile. We provide ShEPhERD with the reference ligand and generate 100 analogs evenly split between 36, 38, 40, 42, 44, 46, 48, 50, 51, 52, 53, 54, 55, 56, 57, 58, 59, 60, 62, 64, 66, 68, 70, 75, and 80 atoms (4 each). To prepare molecules for ShEPhERD conditioning, we generate partial charges for each reference ligand using xTB(Bannwarth et al., 2019). For CGFlow-ZS experiments, we generate molecules in a zero-shot protein-conditioned setting using TacoGFN (Shen et al., 2024) first implemented by Seo et al.(Seo et al., 2024; Shen et al., 2024) Molecules are generated using the web app described in the GitHub repository[4], which inherits the sampling hyperparameters specified in the original CGFlow manuscript. For each target, we conditionally generate using a cleaned PDB and centroid derived from reference ligand heavy atoms. See  Appendix D.6 for details of DiffLinker in linker design experiments.

## D    Extended results and discussion

### D.1    Training Ablations

Table 3: Training ablations. We incrementally remove inference annealing, auxiliary losses, self-conditioning, scaled-noise, and constraints to see the performance difference. All results shown are at 50 epochs rather than 100 epochs in Table 1. Here, "Constraints" refers to both training-time compatibility masking and sampling constraints. See Sections 4.3 and 4.4, (Appendices B.7, B.14 and B.16).

| Method | Valid. ↑ | GFN-FF ↓ |
|---|---|---|
| Base | 93.5 | 4.871 |
| - Inference annealing | 93.5 | 4.933 |
| - Auxiliary losses | 85.3 | 5.194 |
| - Self-conditioning | 69.0 | 6.424 |
| - Scaled noise | 70.4 | 5.091 |
| - Constraints | 42.4 | 67.006 |

### D.2    Sampling Ablations

By default, SYNCOGEN implements a linear noise schedule and samples for 100 timesteps. To evaluate the effect of step count and noise schedule choice on performance, we provide experiments

---

[1]`https://github.com/rssrwn/semla-flow/`, available under the MIT License

[2]`https://github.com/jule-c/eqgat_diff/`, `https://github.com/cvignac/MiDi`, `https://github.com/GRAPH-0/JODO`, `https://github.com/Dunni3/FlowMol`,    available under the MIT License

[3]`https://github.com/wenhao-gao/synformer`

[4]`https://github.com/tsa87/cgflow`

with step count decreased to 50 and 20, as well as modified noising to follow a log-linear and geometric schedule. All results listed subsequently can be assumed to use the default noise schedule and step count.

We additionally follow FoldFlow to implement *inference annealing*, a time-dependent scaling on Euler step size that was found to empirically improve in-silico results in protein design Bose et al. (2024). We studied multiplying the Euler step size at time $t$ by $5t$, $10t$, and $50t$. In practice, we employ $10t$ for our experiments unless otherwise noted.

We find that noising and de-noising building blocks according to a linear noise schedule generally achieves good performance, which during inference sees most unmasking occur in the final steps. An aggressive denoising schedule for the discrete fragments yields significantly worse validity (Geometric and Loglinear). Inference annealing that speeds up continuous denoising in the beginning but slows it down near the end helps to inform discrete unmasking and can slightly improve discrete generation validity, energies, and PoseBusters validity. As a sanity check to evaluate whether simultaneous generation is necessary for good performance using SYNCOGEN, we evaluate an inference configurations where all building blocks and reactions are noised until a single final prediction step (FinalOnly) where we find performance using the default parameters to be superior.

Table 4: Sampling ablations. Results are averaged over 1000 generated samples, except retrosynthesis solve rate (out of 100). All results shown are at 50 epochs rather than 100 epochs in Table 1.

| | **Primary metrics** | | | | | **Secondary metrics** | | |
|---|---|---|---|---|---|---|---|---|
| **Method** | Valid. ↑ | AiZyn. ↑ | Synth. ↑ | GFN-FF ↓ | GFN2-xTB ↓ | PB ↑ | Div. ↑ | Nov. ↑ |
| Linear-100 | 93.5 | 55 | 70 | 4.933 | -0.92 | 78.3 | 0.79 | 94.1 |
| Linear-20 | 82.4 | 56 | 68 | 5.102 | -0.91 | 71.3 | 0.78 | 94.9 |
| Linear-50 | 92.0 | 50 | 65 | 4.890 | -0.91 | 78.9 | 0.78 | 93.6 |
| Geometric-100 | 48.2 | 61 | 68 | 5.206 | -0.84 | 72.0 | 0.80 | 91.7 |
| Loglinear-100 | 60.3 | 56 | 64 | 5.182 | -0.87 | 70.1 | 0.80 | 91.7 |
| Annealing-$5t$ | 94.7 | 52 | 58 | 5.001 | -0.93 | 79.1 | 0.78 | 94.1 |
| **Annealing-$10t$ (default)** | 93.5 | 42 | 68 | 4.870 | -0.91 | 82.8 | 0.78 | 94.2 |
| Annealing-$50t$ | 85.1 | 51 | 64 | 4.972 | -0.82 | 86.7 | 0.76 | 94.6 |
| FinalOnly | 69.7 | 39 | 68 | 5.260 | -0.92 | 70.1 | 0.76 | 94.1 |

To examine the effect of de-noising edges probabilistically ("Default") against exclusively selecting the highest-probability edge ("Argmax") during sampling, we sample 1000 molecules unconditionally for each setting.

Table 5: Comparison of sampling strategies. All results are at 100 epochs (same with Table 1) but without inference annealing.

| **Sampling Mode** | **Validity ↑** | **PB ↑** | **Diversity ↑** | **Novelty (%) ↑** |
|---|---|---|---|---|
| Default | 0.947 | 0.8425 | 0.7811 | 95.24 |
| Argmax | 0.956 | 0.8483 | 0.7797 | 93.41 |

We see there is relatively insignificant differences (small improvement in the proportion of valid molecules is slightly higher in the "Argmax" setting, at the cost of slightly lower diversity and novelty). Note that the proportion of molecules with 3, 4 and 5 building blocks are sampled according to their respective distributions in SynSpace.

### D.3 LARGER VOCABULARY

Appendix D.3 show unconditional generation results after training SYNCOGEN on SYNSPACE-Lfor 50 epochs, and as compared to SYNCOGEN on SYNSPACE for 50 epochs. We note that given a thousand fold increase in search space, longer training would typically be expected, and may further improve the results. Under this conservative setting, scaling to the larger search space yields very similar overall behavior. We observe a small decrease in RDKit validity and a moderate decrease in PoseBusters validity, and the molecules still retain good conformer energies and high

retrosynthesis solve rates. These indicate the molecules are synthesizable, and the local geometry remains reasonable. Diversity is essentially unchanged, while novelty increases substantially from 94.2% to 99.1%, showing that the enlarged vocabulary is effectively used to explore new regions of chemical space.

| Method | Primary metrics | | | | | Secondary metrics | | |
|---|---|---|---|---|---|---|---|---|
| | Valid. ↑ | AiZyn. ↑ | Synth. ↑ | GFN-FF ↓ | GFN2-xTB ↓ | PB ↑ | Div. ↑ | Nov. ↑ |
| SYNCOGEN SYNSPACE | 93.5 | 42 | 68 | 4.870 | -0.91 | 82.8 | 0.78 | 94.2 |
| SYNCOGEN SYNSPACE-L | 87.0 | 52 | 77 | 5.502 | -0.81 | 65.0 | 0.79 | 99.1 |

## D.4 METRICS

We here describe metric computation details that are absent in the main text.

For synthesizability evaluation, we used the public AiZynthFinder and Syntheseus models. Due to the speed of these models, we only evaluate 100 randomly sampled generated examples. For AiZynthFinder, we used the USPTO policy, the Zinc stock, and we extended the search time to 800 seconds with an iteration limit of 200 seconds. For Syntheseus, we used the LocalRetro model with Retro* search under default settings, with Enamine REAL strict fragments as the stock. We additionally appended our building blocks as the stock, but found no meaningful difference in solved rates, presumably as most of our building blocks are already in the utilized stock. We note that we replaced all boranes with boronic acids due to simplifications made in our modeling (see Appendix A.2).

For energy evaluation, all results are from single-point calculations. For GFN-FF, we report the total energy minus the bond energies (equivalent to the sum of angle, dihedral, bond repulsion, electrostatic, dispersion, hydrogen bond, and halogen bond energies) as the intramolecular non-bond energies, and average it over the number of atoms. For GFN2-xTB, we report the dispersion interaction energies as the intramolecular non-covalent energies. We note that the total energies and bonded energies follow very similar trends. We note that MMFF94 energies are not parameterized for boron; therefore, we report them only for the Wasserstein distances in Appendix D.5 and inpainting task in Table 8. Figures 3 and 14 show distributions obtained from 1,000 molecules generated by each generative method, along with 50,000 subsampled molecules from their respective training datasets. Gaussian kernel density estimation (bandwidth = 0.15) was used for linear distributions, while von Mises kernel density estimation ($\kappa = 25$) was applied for circular distributions. Wasserstein-1 distances (computed linearly for lengths and energies, and on the circle for angles and dihedrals) were calculated using the Python Optimal Transport Package (Flamary et al., 2021).

## D.5 *De novo* 3D MOLECULE GENERATION

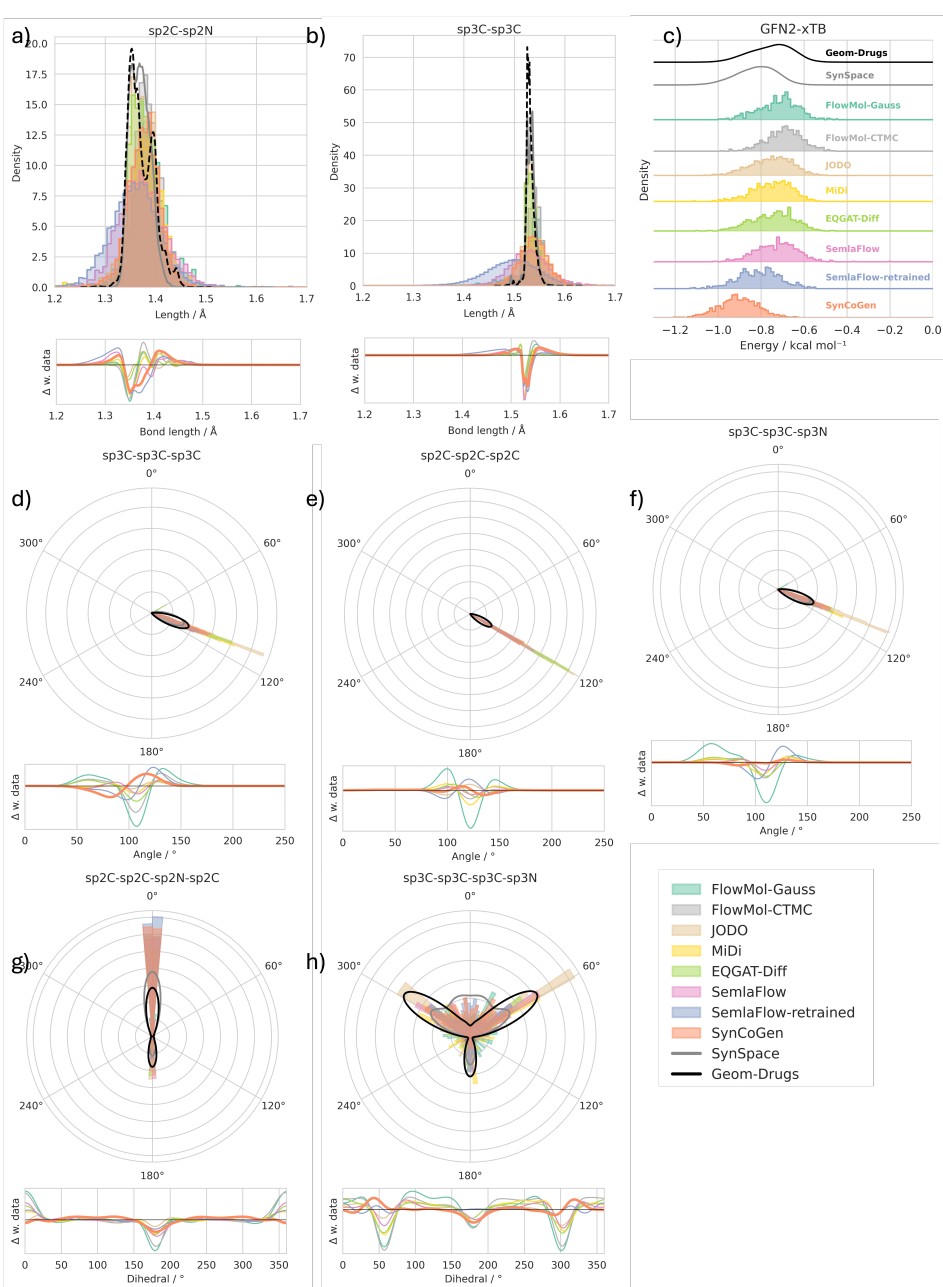

Figure 14: **Additional conformer bond length, angle, dihedral, and energy distribution comparisons.** a-b) Bond lengths, c) GFN2-xTB energy distribution, d-f) bond angles, g-h) dihedral angles. Solid curves denote training data densities; lower subpanels show deviations between generated samples and data.

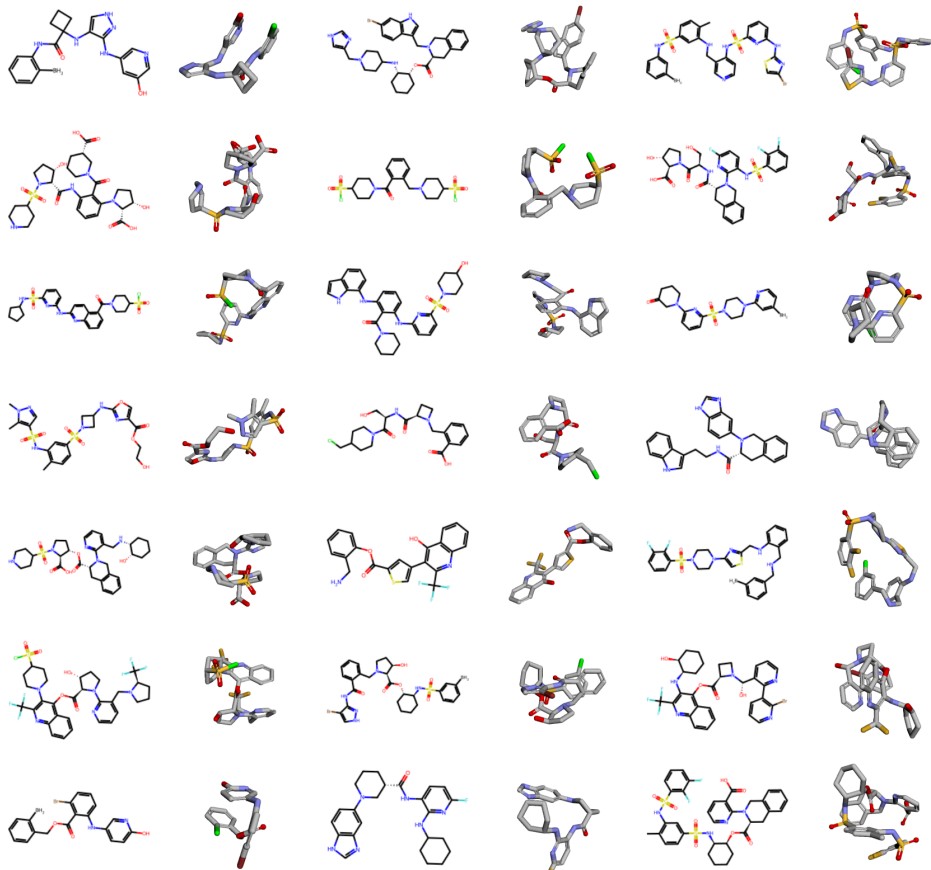

Figure 15: Unconditionally sampled random molecules from SYNCOGEN.

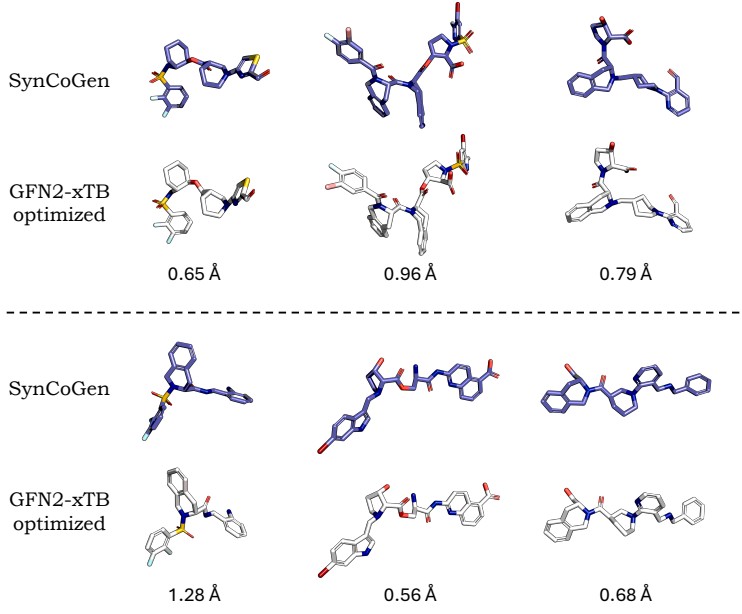

Figure 16: A subset of randomly sampled molecules from SYNCOGEN and further optimized by GFN2-xTB until convergence. Alignment RMSD is shown below the molecular structures.

Table 6: Wasserstein-1 distance ($W_1$) and Jensen–Shannon divergence (JSD) for the generative models (lower is better). For bond lengths, angles, and dihedrals, we computed the average $W_1$ and JSD for the top 10 prevalent lengths/angles/dihedrals. Comparisons are made to the respective training set.

(a) Bond dihedrals

| Method | $W_1$ | JSD |
|---|---|---|
| SYNCOGEN | 7.01 | 0.29 |
| SEMLAFLOW SYNSPACE | 6.50 | 0.22 |
| SEMLAFLOW | 7.76 | 0.28 |
| EQGAT-Diff | 8.48 | 0.29 |
| MiDi | 9.32 | 0.38 |
| JODO | 5.47 | 0.31 |
| FlowMol-CTMC | 13.69 | 0.35 |
| FlowMol-Gauss | 18.85 | 0.46 |

(b) Bond angles

| Method | $W_1$ | JSD |
|---|---|---|
| SYNCOGEN | 1.36 | 0.22 |
| SEMLAFLOW SYNSPACE | 1.64 | 0.28 |
| SEMLAFLOW | 1.18 | 0.21 |
| EQGAT-Diff | 1.37 | 0.16 |
| MiDi | 1.41 | 0.21 |
| JODO | 0.59 | 0.12 |
| FlowMol-CTMC | 1.90 | 0.24 |
| FlowMol-Gauss | 3.68 | 0.30 |

(c) Bond lengths

| Method | $W_1$ | JSD |
|---|---|---|
| SYNCOGEN | 0.0171 | 0.34 |
| SEMLAFLOW SYNSPACE | 0.0320 | 0.48 |
| SEMLAFLOW | 0.0200 | 0.38 |
| EQGAT-Diff | 0.0039 | 0.13 |
| MiDi | 0.0142 | 0.31 |
| JODO | 0.0034 | 0.12 |
| FlowMol-CTMC | 0.0089 | 0.20 |
| FlowMol-Gauss | 0.0152 | 0.28 |

(d) `GFN2-xTB` non-covalent $E$

| Method | $W_1$ | JSD |
|---|---|---|
| SYNCOGEN | 0.0838 | 0.33 |
| SEMLAFLOW SYNSPACE | 0.0125 | 0.16 |
| SEMLAFLOW | 0.0249 | 0.16 |
| EQGAT-Diff | 0.0073 | 0.12 |
| MiDi | 0.0084 | 0.14 |
| JODO | 0.0031 | 0.11 |
| FlowMol-CTMC | 0.0605 | 0.26 |
| FlowMol-Gauss | 0.0322 | 0.19 |

(e) `GFN-FF` non-bonded $E$

| Method | $W_1$ | JSD |
|---|---|---|
| SYNCOGEN | 1.37 | 0.28 |
| SEMLAFLOW SYNSPACE | 1.09 | 0.22 |
| SEMLAFLOW | 1.52 | 0.16 |
| EQGAT-Diff | 1.69 | 0.18 |
| MiDi | 1.80 | 0.19 |
| JODO | 1.33 | 0.12 |
| FlowMol-CTMC | 1.53 | 0.17 |
| FlowMol-Gauss | 2.13 | 0.17 |

(f) `MMFF` total $E$

| Method | $W_1$ | JSD |
|---|---|---|
| SYNCOGEN | 6.59 | 0.089 |
| SEMLAFLOW SYNSPACE | 54.63 | 0.22 |
| SEMLAFLOW | 69.56 | 0.24 |
| EQGAT-Diff | 4.80 | 0.076 |
| MiDi | 19.00 | 0.11 |
| JODO | 22.07 | 0.11 |
| FlowMol-CTMC | 41.95 | 0.15 |
| FlowMol-Gauss | 26.96 | 0.14 |

Table 7: With given reaction graphs, comparison of mean coverage (COV) and matching accuracy (MAT) for RDKit ETKDG and zero-shot conformer generation using SYNCOGEN.

| Method | COV (%) ↑ | MAT (Å) ↓ |
|---|---|---|
| RDKit | 0.692 | 0.657 |
| SYNCOGEN | 0.614 | 0.693 |

### D.6 MOLECULAR INPAINTING EXPERIMENTS

Three protein–ligand complexes (PDB IDs 7N7X[5], 5L2S[6] and 4EYR[7]) were selected for molecular inpainting of the ligand structures. These ligands were chosen because they are prominent FDA-approved drugs, and they are typically challenging to synthesize, but the key functional groups are present in our building blocks. Specifically, 4EYR contains ritonavir, a prominent HIV protease inhibitor on the World Health Organization's List of Essential Medicines; 5L2S contains abemaciclib, an anti-cancer kinase inhibitor that is amongst the largest selling small molecule drugs; 7N7X contains berotralstat, a recently approved drug that prevents hereditary angioedema. Note that for 4EYR, the inpainting was done using the ligand geometry from the PDB entry 3NDX[8], but docking was performed with 4EYR because the protein structure in 3NDX contained issues – nonetheless, both entries contain the same protease and ligand.

In addition to the experiments in Section 5.2, we evaluate SYNCOGEN's conditional sampling performance for the fragment linking framework against the state-of-the-art model DiffLinker (Igashov et al., 2024). While DiffLinker is trained for fragment-linking, our model performs zero-shot fragment linking without any finetuning. For both models, the size of the linker was chosen so that it matches

---

[5] https://www.rcsb.org/structure/7N7X
[6] https://www.rcsb.org/structure/5L2S
[7] https://www.rcsb.org/structure/4EYR
[8] https://www.rcsb.org/structure/3NDX

that of the original ligand: 2 extra nodes were sampled for SYNCOGEN and 15 linking atoms for DiffLinker in the case of 5L2S, while 3 extra nodes and 25 linking atoms were sampled for 4EYR and 7N7X. We specified leaving groups (for SYNCOGEN) and anchor points (for DiffLinker) so that the fragments are linked at the same positions as in the ligand. Results are shown in Table 8. No retrosynthetic pathways were found for the molecules in DiffLinker, while SYNCOGEN models synthetic pathways and synthetic pathways can be easily drawn, with examples for 4EYR shown in Figure 18. This out-of-distribution task for SYNCOGEN leads to fewer valid molecules; however, for the valid candidates, SYNCOGEN has lower interaction energies and achieves 100% connectivity as it uses reaction-based assembly, whereas DiffLinker can sample disconnected fragments.

Table 8: Molecular inpainting task. Results are averaged over 1000 generated samples, except retrosynthesis solve rate (out of 100). SYNCOGEN-FT denotes a light fine-tuning model for 5 epochs on in-painting with randomly fixed fragments from SYNSPACE.

| Method | Target | AiZyn. ↑ | Synth. ↑ | Valid. ↑ | Connect. ↑ | MMFF ↓ | GFN-FF ↓ | GFN2-xTB ↓ | Diversity ↑ | PB ↑ |
|---|---|---|---|---|---|---|---|---|---|---|
| DiffLinker | 5L2S | 0 | 0 | 95.8 | 95.09 | 14.22 | 7.52 | -0.95 | 0.60 | 49.3 |
| | 4EYR | 0 | 0 | 93.7 | 81.86 | 20.01 | 8.49 | -1.03 | 0.81 | 35.0 |
| | 7N7X | 0 | 0 | 95.8 | 74.65 | 20.51 | 7.99 | -1.09 | 0.78 | 37.5 |
| SYNCOGEN | 5L2S | 73 | 79 | 57.6 | 100 | 10.11 | 6.77 | -0.78 | 0.62 | 27.3 |
| | 4EYR | 72 | 58 | 46.9 | 100 | 12.80 | 6.58 | -0.86 | 0.64 | 32.0 |
| | 7N7X | 53 | 69 | 50.6 | 100 | 4.243 | 6.60 | -0.80 | 0.67 | 56.1 |
| SYNCOGEN-FT | 5L2S | 77 | 84 | 75.3 | 100 | 4.25 | 6.58 | -0.81 | 0.632 | 56.2 |
| | 4EYR | 42 | 78 | 62.0 | 100 | 10.13 | 5.33 | -0.78 | 0.604 | 19.8 |
| | 7N7X | 57 | 77 | 73.6 | 100 | 4.09 | 6.86 | -0.83 | 0.664 | 47.9 |

Table 9: Percentage of hard-to-synthesize chemical features in generated "valid" molecules from SYNCOGEN versus DiffLinker in fragment linking (out of 1000). Exotic bonds include hydrazine, nitro, nitramine, azide, diazo, peroxide, nitrate ester, fulminate. Fused large/small rings are where a fused ring contains a sub-ring that is larger than 6 atoms or smaller than 5 atoms.

| | DiffLinker | | | SYNCOGEN | | |
|---|---|---|---|---|---|---|
| Chemical features | 5L2S | 3NDX | 7N7X | 5L2S | 3NDX | 7N7X |
| Macrocycles (>=9) | 1.0% | 72.6% | 12.6% | **0.0%** | **0.0%** | **0.0%** |
| Fused rings with large/small rings | 13.3% | 81.4% | 37.0% | **0.0%** | **0.0%** | **0.0%** |
| Large rings (7,8) | 12.1% | 9.9% | 22.2% | **0.0%** | **0.0%** | **0.0%** |
| Disconnected | 4.9% | 18.1% | 25.4% | **0.0%** | **0.0%** | **0.0%** |
| Exotic bonds | 0.2% | 1.2% | 1.7% | **0.0%** | **0.0%** | **0.0%** |
| Total problematic % | 22.8% | 86.0% | 61.5% | **0.0%** | **0.0%** | **0.0%** |

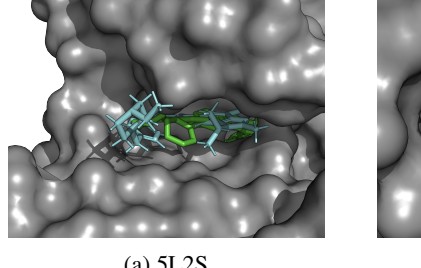 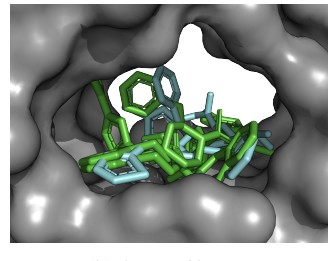 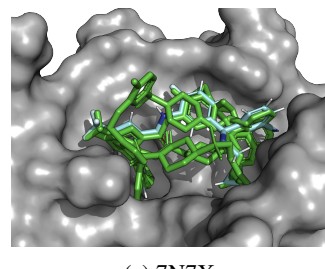

|  |  |  |
|---|---|---|
| (a) 5L2S | (b) 3NDX/4EYR | (c) 7N7X |

Figure 17: Structural overlays of the native protein (gray) and its native ligand (blue) with AlphaFold3-predicted folds of a subset of generated ligands (green) for (a) 5L2S, (b) 3NDX/4EYR, and (c) 7N7X.

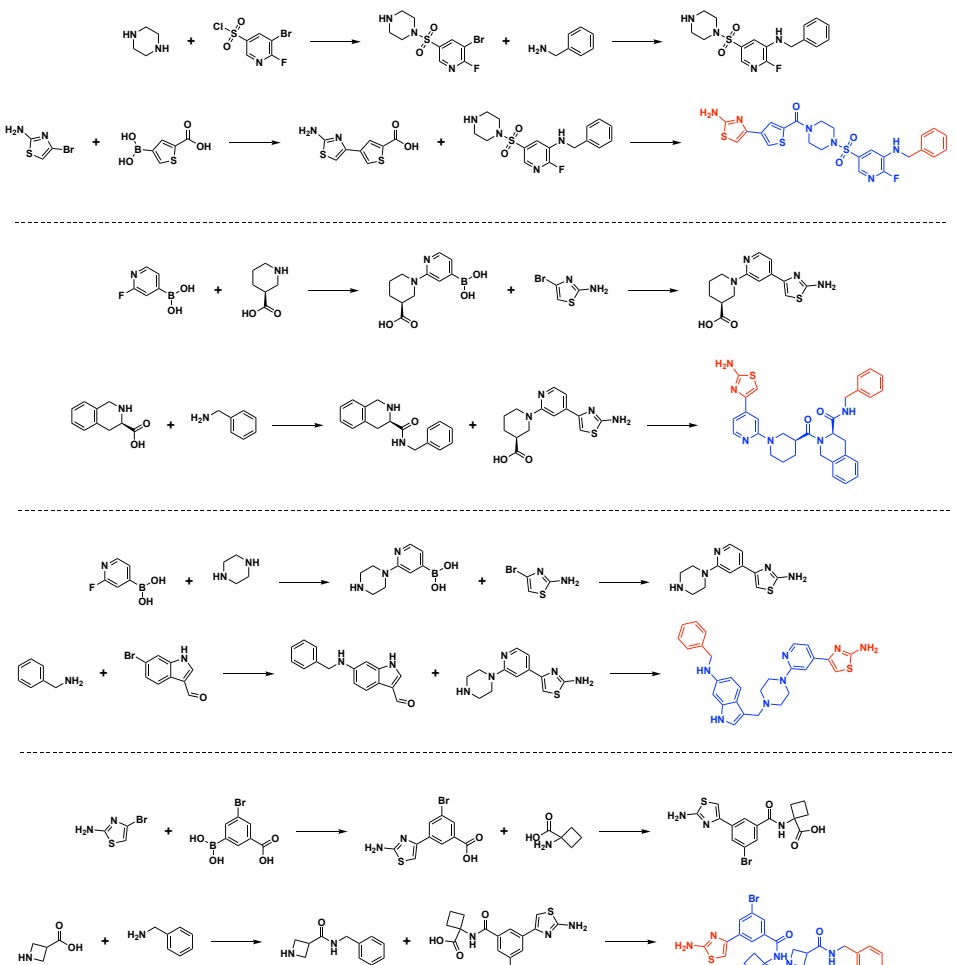

Figure 18: Synthetic pathways for molecules generated in the molecular inpainting task for target 3NDX/4EYR. The final product is shown in blue, and the inpainted fragments are shown in red.

## D.7 PHARMACOPHORE-CONDITIONED GENERATION EXPERIMENTS

Table 10: Percentage of hard-to-synthesize chemical features in pharmacophore generation for CGFlow-ZS, Shepherd, Synformer, and SYNCOGEN 100 per target, 10 targets in total). Exotic bonds include hydrazine, nitro, nitramine, azide, diazo, peroxide, nitrate ester, fulminate. Fused large/small rings are where a fused ring contains a sub-ring that is larger than 6 atoms or smaller than 5 atoms.

| Chemical features | CGFlow-ZS | Shepherd | Synformer | SYNCOGEN |
|---|---|---|---|---|
| Macrocycles (>=9) | **0.0%** | 4.7% | 1.8% | **0.0%** |
| Fused rings with large/small rings | 31.1% | 39.2% | 1.9% | **0.1%** |
| Large rings (7,8) | 1.2% | 31.2% | 4.0% | **0.0%** |
| Disconnected | **0.0%** | **0.0%** | **0.0%** | **0.0%** |
| Exotic bonds | **0.0%** | 0.3% | 1.3% | **0.0%** |
| Total problematic % | 31.3% | 46.8% | 8.3% | **0.1%** |

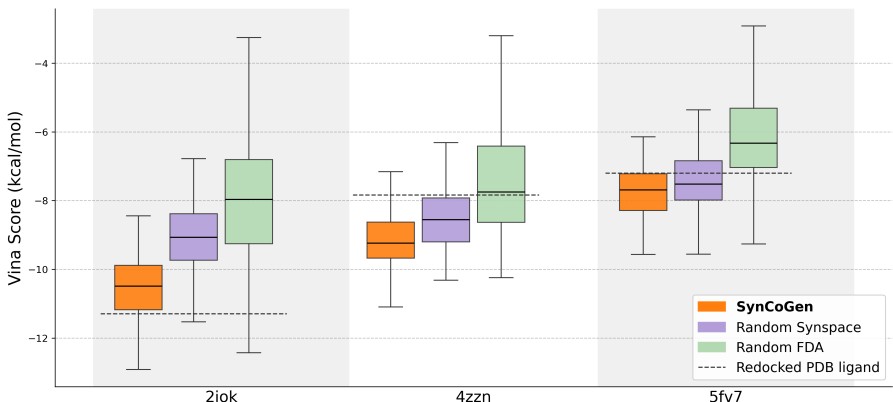

Figure 19: Docking score box-plot comparisons on pharmacophore-conditioned SYNCOGEN samples, randomly selected SYNSPACE samples, and randomly selected FDA-approved small molecules (100 for each target). Pharmacophore-conditioned SYNCOGEN outperforms SYNSPACE, which outperforms FDA-approved molecules. These results suggest that the reason why pharmacophore-conditioned SYNCOGEN can outperform other baselines may partially stem from the careful curation of building blocks, as SYNSPACE samples perform well in docking experiments. Lastly, we caution that docking is a merely a proxy for binding affinity, and we emphasize that the primary results are that SYNCOGEN generates synthesizable molecules with reasonable poses when conditioned on pharmacophore profiles. Note all SYNCOGEN sampling runs were performed using a building block count fixed to 3.

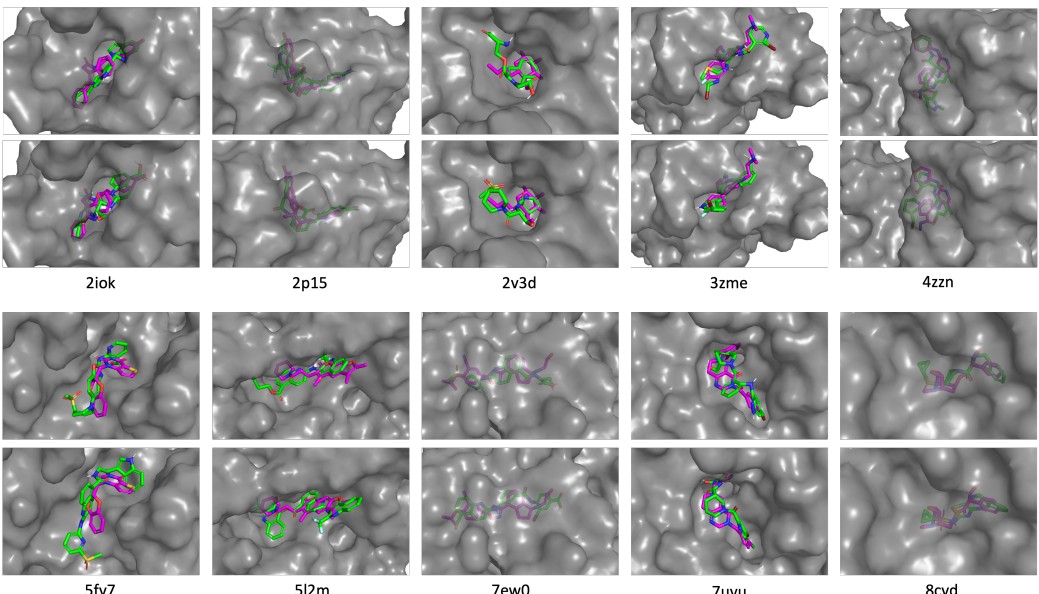

Figure 20: **Pharmacophore-conditioning task.** Examples of docked SYNCOGEN-generated molecules (green) overlaid with PDB ligands (magenta) in their crystal structure pose.

