# OpenReview forum: "SynCoGen: Synthesizable 3D Molecule Generation via Joint Reaction and Coordinate Modeling"
_ICLR.cc/2026/Conference — ICLR 2026 Poster_

### Official Review · Reviewer_t7Yp · 2025-10-29

**Soundness:** 3
**Presentation:** 2
**Contribution:** 3
**Rating:** 6
**Confidence:** 4

**Summary:**

This paper introduces SynCoGen, a framework that jointly generates synthesizable reaction graphs and 3D molecular coordinates using masked discrete diffusion and flow matching. The paper also introduces SynSpace, a new dataset with 600k synthesis-aware molecules. Experiments show strong results in unconditional 3D generation, fragment linking, and pharmacophore-conditioned tasks.

**Strengths:**

1. Novel idea of coupling synthesizability with 3D coordinate generation.
2. The dataset is well-curated and likely useful for future work.
3. Results are convincing and demonstrate clear improvements over baselines.

**Weaknesses:**

1. There are many node/edge modeling constraints, heavy notation, and complex noise processes that make the method difficult to follow and reproduce. It’s also unclear which components contribute most to the final performance — the paper did not provide sufficient ablation studies or analysis isolating the impact of individual design choices (e.g., compatibility masking, visibility-aware noise, joint time coupling). As a result, the practical necessity of the full system remains uncertain.
2. The reaction vocabulary is fixed and limits diversity. The model relies on a predefined set of 93 building blocks and 19 reaction templates, where each reaction involves at most one leaving group per reagent. While this makes synthesis simulation simpler and chemically reliable, it also restricts the model’s capacity to explore more diverse or complex chemical spaces.

**Questions:**

1. How scalable is SynCoGen to larger or more diverse reaction vocabularies? Would the framework still function if reactions involved multiple leaving groups or more complex coupling patterns?
2. The paper claims unified discrete–continuous time coupling. However, it seems that diffusion or flow updates on edge types may not take effect until the corresponding nodes are decoded. Did the authors experiment with decoupled or asynchronous schedules across modalities, and if so, how did that affect performance?

I would consider raising my scores if authors could give some resonable explanations to the questions.

---

> ### Author Response · Authors · 2025-11-24
> **Response - reviewer t7Yp (1/2)**
>
> We thank the reviewer for the thoughtful comments and for highlighting the novelty, the dataset, and the strong empirical results. We address each concern in turn.
>
> ### Weakness 1
> We thank the reviewer for the feedback and agree that the method section could benefit from diagrams explaining the workflow. We **added step-by-step diagrams and corresponding explanations** in Appendices B.1, B.2, and B.3 in the revised manuscript. We have also **updated Section 4.2**. These diagrams, at a glance, illustrate (i) how masked graphs and coordinates are noised, (ii) how compatibility masks and visibility masks are applied, and (iii) how the joint sampler updates $(X, E, C)$ at each time step.
>
> Appendix D.1 now reports explicit training ablations where we progressively remove key components (see Overall Response, “Method clarifications”). We show that auxiliary losses and scaled noise have a small impact on performance, whereas constraints (both compatibility constraints and sampling-time edge constraints) and self-conditioning (see Appendix D.1) are critical.
>
> We did attempt to remove the visibility/padding mechanism and instead train with “true” atom counts only. This created a mismatch between training (known atom counts per block) and sampling (unknown atom counts whenever a block is masked). Empirically, this led to severe mode collapse in the building-block usage and poor coordinate quality, so we now treat **visibility masking as part of the core architecture** and explain this explicitly in Appendices B.1 and B.6.
>
> Overall, we hope this combination of diagrams and quantitative ablations clarifies which components are essential and why.
>
> ### Weakness 2
> We thank the reviewer for the comment, please see the **“Search space and dataset analysis” section of the Overall Response** and response to reviewer 8SMJ's W1/Q1 for our detailed response. In summary, the curated building blocks and reactions were deliberately chosen for robustness and availability, but the induced space is still large: SynSpace contains >600k molecules and ~4.8× more unique Murcko scaffolds than GEOM-Drugs (443k vs 93k), while remaining in a realistic lead-like region (higher sp³ fraction, TPSA, rotatable bonds). This suggests that, despite the fixed vocabulary, the model explores a chemically rich space with practical synthetic routes. As our experiments suggest, training on SynSpace yields competitive results to baseline methods, highlighting that the **vocabulary does not limit performance** (even on OOD conditional generation tasks, see Section 5.3). We note that the architecture allows for arbitrarily large building-block libraries, and we are already curating an expanded library (e.g., ~378 building blocks and 26 reactions). Finally, we will release a script that regenerates SynSpace-style datasets from arbitrary user-defined block/template sets.
>
> ### Question 1
> The model itself is straightforwardly compatible with more leaving groups and more complicated couplings. However, reactions that involve ring formation introduce complexity that requires custom atom attribution between reactants and products; RDKit does not track this when simulating chemical reactions. We now clarify this in Appendix A.2.
>
> ### Question 2
> We would like to clarify that SynCoGen does **not** require building blocks to be denoised before edges. Both nodes and edges are denoised under the same time-dependent unmasking probabilities determined by the discrete noise schedule. We now clarify this in Figure 1’s caption.
>
> To directly test the reviewer’s suggestion on decoupled/asynchronous schedules, we provide ablations over discrete noise schedules in Appendix D.2, ranging from geometric (high early-time transition probabilities) to log-linear (transition probabilities increasing linearly in time) to linear (transition probabilities increasing exponentially in time). Note that these are set independently of the coordinate interpolation, which we fix to linear for these experiments.
>
> We find that a linear discrete noise schedule yields optimal performance, whereas aggressive early unmasking of fragments substantially __hurts validity__. Furthermore, the “FinalOnly” configuration, where all nodes/edges are forced to remain masked until a single final step, also shows significantly worsened performance.
>
> For coordinate interpolation, we find empirically that inflating the linear Euler step size early during sampling by multiplying by $beta$ $t$, where $beta$ is some coefficient and $t$ is the time step, improves energies. This corroborates the results obtained by Huguet et al.[1] in applying the same strategy.
>
> Taken together, these results show that the a naive decoupled/asynchronous scheme performs markedly worse, and the **specific joint schedule we use is important for stable co-generation**.

---

> ### Author Response · Authors · 2025-11-24
> **Response - reviewer t7Yp (2/2)**
>
> ### Concluding remarks
> We appreciate the reviewer’s focus on interpretability and the necessity of our design choices. The revised manuscript now (i) has an extended discussion on the impact of constraints, self-conditioning, and joint schedules, (ii) clarifies why visibility-aware masking is needed, and (iii) explains the choice of the building blocks/reactions as well as how the framework may scale to richer reaction vocabularies. We hope these additions address the reviewer’s concerns and would be very grateful if the reviewer could reconsider the score in light of the strengthened evidence.
>
> [1] Huguet, G., Vuckovic, J., Fatras, K., Thibodeau-Laufer, E., Lemos, P., Islam, R., … Bose, A. J. (2024). Sequence-Augmented SE(3)-Flow Matching For Conditional Protein Backbone Generation. arXiv [Cs.LG]. Retrieved from http://arxiv.org/abs/2405.20313

---

### Official Review · Reviewer_1Mne · 2025-10-31

**Soundness:** 3
**Presentation:** 1
**Contribution:** 3
**Rating:** 4
**Confidence:** 4

**Summary:**

The authors introduce.  SYNCOGEN, a multimodal generative framework that jointly models 3D molecular geometry and synthetic accessibility through reaction templates. The key idea is to integrate masked graph diffusion and flow matching into a unified generative process that simultaneously samples both the reaction graph over molecular building blocks and the corresponding atomic coordinates. This coupling enables SYNCOGEN to generate 3D molecules that are not only conformationally realistic but also chemically synthesizable via known reaction templates. To support this framework, the authors curate SYNSPACE, a large-scale dataset of over 600K synthesizable molecules and 3.3M associated low-energy conformations, represented as building-block reaction graphs. The model supports conditional generation tasks such as linker design and pharmacophore-guided generation, showcasing its utility in practical drug discovery.

**Strengths:**

The paper tackles one of the central challenges in molecular generative models i.e. providing accessible synthetic routes for generated candidates. Additionally the proposed method is able to generate favourable conformers.

**Weaknesses:**

- The writing of the paper is convoluted makes it difficult to follow and appreciate the methodology. It would be beneficial if the authors explain their method more clearly. For example, there are several components in the proposed workflow, a better workflow diagram explaining end-to-end working of SYNCOGEN will be helpful to the readers.
- for a complicated method as yours, with so many components, a more detailed workflow will he helpful. The message in current fig1 is unclear
- To better understand the method, code could have helped. However the url provided (https://anonymous.4open.science/r/SynCoGen-13F7) does not have the code. Please provide reproducible code.
- Another major issue I have that authors should benchmark their work against Shen et al (2025). It is the closest baseline to their work however haven't been benchmark against. I understand Shen et al, is conditional SBDD generation but their attempt to generate conformers while simultaneously providing reaction pathway is most similar to this work. In theory, the authors unconditional generation should be able to outperform Shen et al. Please include this experiment in your paper.

**Questions:**

- Fig 1 caption, what does non-linear node in this context mean?
- why do authors disallow macrocycles?

---

> ### Author Response · Authors · 2025-11-24
> **Response - reviewer 1Mne**
>
> We thank the reviewer for carefully reading the paper and providing constructive feedback, and also for emphasizing the importance of simultaneously addressing synthetic accessibility and 3D conformer quality. We now address each of the reviewer's points below.
>
> ### Weakness 1 & 2
> We agree that the method section was too dense. In the revision, we have added **detailed figures and corresponding explanations in Appendices B.1, B.2, and B.3.** These now step the reader through how we perform training, highlighting ground truth preparation and index masking, which are bespoke to our setup. We also added a diagram that details compatibility logit masking for building blocks and the strategy we employ in sampling for edge logit masking. We have also **reworked Section 4.2**. We sincerely thank the reviewer for pointing this out and believe that these changes now facilitate a better understanding of our methods.
>
> ### Weakness 3
> We apologize for the broken link. We have uploaded updated anonymized code as supplemental information. In addition, we now release SynSpace, pharmacophores, and trained unconditional and pharmacophore-conditioned weights at the URL detailed in the anonymized code README.
>
> ### Weakness 4
> We thank the reviewer for the suggestion and agree that this comparison is necessary. As mentioned in the **“Stronger baselines” section of the Overall Response**, we benchmark the model directly against zero-shot CGFlow generation.
>
> To make the comparison fair, we use a zero-shot, amortized version of CGFlow (CGFlow-ZS) that shares our setting: one model, no per-target retraining, pocket-conditioned reward only at sampling. This avoids comparing our amortized approach against separate retraining for each target that specifically optimizes for docking rewards and is very expensive. Our results show that SynCoGen’s top samples outperform every baseline in docking on 8 out of 10 targets, and achieve higher synthesizability with fewer challenging motifs (macrocycles, exotic fused rings) than CGFlow.
>
> ### Question 1
> We apologize for the typo. The caption now reads “Note that **molecules** are not necessarily linear…”. We meant that there is no assumption of a linear reaction graph. This has now been clarified in the revised text.
>
> ### Question 2
> Our design goal in this work is a high-throughput, building-block–based synthesis regime that is easy to execute on standard parallel chemistry platforms. Macrocycles constitute a class of structures that are typically challenging to synthesize at scale and as such do not quite align with our design objective of focusing on synthetically accessible building-block assemblies. We added a note in Section 4.3. Despite this, the SynSpace dataset demonstrates significant coverage of chemical space, and we show that the model is able to generate analogs to a diverse array of reference ligands, as discussed in the Overall Response.
>
>
> ### Concluding remarks
> We are grateful for the detailed suggestions, which directly shaped the new figures, code release, and CGFlow comparison. We hope these additions about clarity, baselines, and scope address the great points raised by the reviewers, and we would greatly appreciate if the reviewer could reconsider the score in light of the strengthened revision. We are of course happy to provide any further details or analyses that might be helpful.

---

### Official Review · Reviewer_7GM3 · 2025-11-01

**Soundness:** 2
**Presentation:** 3
**Contribution:** 3
**Rating:** 4
**Confidence:** 4

**Summary:**

This paper introduces SYNCOGEN, a generative framework for designing synthesizable small molecules directly in 3D Cartesian space. The core idea is to jointly model molecular building blocks and chemical reactions alongside the continuous 3D atomic coordinates. The method combines a masked graph diffusion model for the reaction graph with a flow matching model for the coordinates, trained on a newly curated dataset called SYNSPACE. The authors demonstrate SYNCOGEN's capabilities through experiments in unconditional generation, where it achieves state-of-the-art performance in both 3D geometry and synthesizability metrics, as well as in conditional tasks like fragment linking and pharmacophore-conditioned generation, where it shows competitive results.

**Strengths:**

1. The primary contribution is the novel formulation of jointly generating a synthesizability-aware reaction graph and the corresponding 3D molecular conformation. This elegantly bridges two previously disconnected lines of research: 2D synthesizable generation and 3D all-atom generation. If successful, this approach could significantly accelerate drug discovery pipelines by producing spatially-aware designs that are also practically achievable in the lab.
2. The technical execution is of high quality. The choice to combine masked discrete diffusion for the building block graph and equivariant flow matching for the 3D coordinates is a sophisticated and well-suited generative paradigm for this multimodal task. Furthermore, the curation of the SYNSPACE dataset, which explicitly links reaction graphs with low-energy 3D conformers, is a valuable resource contribution that could enable future research in this direction.
3. The paper is well-written, clearly structured, and generally easy to follow. The methodology is explained with sufficient detail, and Figure 1 provides an excellent conceptual overview of the generative process. The experimental setup and evaluation metrics are well-defined.

**Weaknesses:**

1. While the creation of SYNSPACE is a commendable effort, its presentation as a key contribution is not fully supported by an analysis within the paper. The manuscript lacks a characterization of the dataset's properties. To understand the model's behavior and potential biases, it is crucial to see distributions of key molecular properties (e.g., molecular weight, logP, scaffold diversity) and how they compare to standard benchmarks like ZINC or GEOM-Drugs. It is difficult to assess the complexity and chemical scope of the space the model has learned.
2. The de novo generation experiments (Table 1) compare SYNCOGEN only against all-atom 3D generative models. While this is a necessary comparison for evaluating 3D conformer quality, it omits a critical class of baselines: state-of-the-art synthesizable generators that operate on fragments in 2D or 3D.  Does the added complexity of generating 3D coordinates come at the cost of performance on the core task of synthesizable graph generation? This comparison is needed to fully contextualize the paper's contribution.
3. A central motivation of the paper is the advantage of generating 3D coordinates alongside the synthesis plan. However, the experiments do not provide a clear, causal link demonstrating this advantage. The work successfully shows that it can be done, but not why it is better than alternative pipelines. For synthesizability alone, 2D methods suffice.
4. The evaluation for pharmacophore-conditioned generation is performed on only three target proteins. This small sample size is insufficient to establish statistical significance or to claim general applicability. The results are promising but can be treated as anecdotal case studies rather than a robust validation of the method's capability.
5. The model generates molecules conditioned on a pharmacophore without any information about the target protein pocket. It is therefore surprising that it produces molecules with docking scores superior to the native ligand and other baselines. A deeper analysis is needed to convince the reader that this is not a coincidental finding.

**Questions:**

1. Could you please add an analysis of the SYNSPACE dataset's chemical property distributions to the appendix? This would help in understanding the chemical space your model operates in.
2. Could you comment on the feasibility of adding an experiment that explicitly highlights the synergy of your joint approach? For example, a task involving conditioning on a 3D property where a 2D-first pipeline would likely fail.
3. Could you comment on the generalizability of your findings, given the small number of targets?

---

> ### Author Response · Authors · 2025-11-24
> **Response - reviewer 7GM3**
>
> We thank the reviewer for their detailed and constructive feedback. We are glad to hear the reviewer found the joint generation to be a “novel formulation” and the “technical execution is of high quality,” and that “the paper is well-written, clearly structured.” We now address all the raised points below:
>
> ### Weakness 1 and Question 1
> We fully agree that dataset characterization is essential. In the revision we have added Appendix A.3 and revised Section 3.1, which report detailed distributions of SynSpace properties (MW, logP, QED, TPSA, rotatable bonds, sp³ fraction, HBD/HBA, SAScore) and compare them directly to GEOM-Drugs (Table 2, Fig. 8). In summary, SynSpace occupies a more lead-like, polar, high-sp³ region and, despite the curated vocabulary, contains **~4.8× more unique Murcko scaffolds (443k vs 93k)**. Please see details in our Overall Response, under “Search space and dataset analysis.”
>
> ### Weakness 2 & 3 / Question 2
> We agree that additional comparisons against 2D and 3D synthesizable generators help bolster the results and significance of the paper. In the “Stronger baselines” section of the Overall Response, we detail additional baselines and how SynCoGen achieves **better or competitive docking scores while outperforming all baselines on synthesizability metrics**.
>
> We understand that 2D generators, by construction, guarantee molecular validity. SynCoGen produces >95% valid molecules in unconditional generation and ~85% molecular validity in pharmacophore conditioning. Furthermore, the **restrosynthesis solve rates are significantly higher for SynCoGen than SynFormer/CGFlow** due to the careful curation of building blocks and reactions, and thus we argue adding explicit 3D generation does **not** come at the cost of synthesizable graph quality.
>
> Regarding the __advantage__ of joint 3D + synthesis generation, we note that pharmacophore-conditioned design and fragment linking are intrinsically 3D tasks. Current 2D synthesis-aware molecule generators either need to work through 2D similarity (e.g., SynFormer), or be __retrained__ on a target-by-target basis (e.g. with docking reward). SynCoGen, **a single amortized model**, can match such 3D information and achieve better docking and retrosynthesis than these specialized pipelines, while avoiding reward hacking and significantly reducing computational overhead.
>
> Concisely, SynCoGen is, to our knowledge, the strongest model that is capable of both following tasks:
>
> 1. Zero-shot inference in 3D conditioned on an arbitrary number of reference ligands without re-training, which we demonstrated via phramacophore-conditioning
> 2. Synthesizability-aware building block and reaction-based generation
>
> The “Expanded pharmacophore experiments” in the Overall Response are designed to verify these claims.
>
> ### Weakness 4 and Question 3
> We agree that more targets were needed to draw strong conclusions. The revised paper now evaluates **10 targets**: three disease-relevant systems with challenging ligands and seven additional LIT-PCBA targets. SynCoGen’s performance trends (competitive/better docking, higher retrosynthesis success, fewer undesirable motifs) are **consistent across targets**, which we believe provides substantially stronger evidence that the method generalizes beyond a few hand-picked cases. Please see further details in the Overall Response.
>
> ### Weakness 5
>
> The superior docking scores are indeed intriguing, and we believe they are explained by the **bias of our building blocks towards favorable docking scores**. We tested this hypothesis by randomly docking 100 molecules from SynSpace for three targets, and we compared these to the docking scores of 100 randomly sampled FDA-approved drugs and 100 molecules generated by SynCoGen (Figure 17, Appendix D.6). This analysis suggests that (i) our curated building-block/reaction space favors ligands that dock well, and (ii) pharmacophore conditioning + 3D generation add a real signal on top of that prior. Lastly, we explicitly caution in the text that docking is only a proxy, and our main claim is that SynCoGen can generate synthesizable molecules and their plausible poses with 3D pharmacophore conditioning.
>
>
>
> ### Concluding remarks
> We thank the reviewer again for their time and constructive feedback. We hope that the additional results and clarifications provided in the rebuttal address the concerns raised. If so, we would be encouraged if the reviewer could consider raising the score in light of this new information. We are of course happy to provide any further details or analyses that might be helpful.

---

### Official Review · Reviewer_8SMJ · 2025-11-03

**Soundness:** 3
**Presentation:** 2
**Contribution:** 3
**Rating:** 6
**Confidence:** 3

**Summary:**

The paper introduces a multimodal generative framework that jointly samples a reaction graph with masked graph diffusion and 3D atomic coordinates with flow matching under a unified time schedule. To enable training, the authors curate SYNSPACE, comprising synthesis-aware building-block graphs with associated low-energy conformations. Empirically, SYNCOGEN achieves strong validity and retrosynthetic solve rates while producing low-energy, PoseBusters-plausible conformers. It further demonstrates fragment linking and pharmacophore-conditioned generation with competitive docking scores and markedly higher retrosynthesis success than baselines.

**Strengths:**

- Well-motivated co-generation: Jointly models reaction graphs and 3D coordinates, explicitly aligning “makeability” with structural plausibility in one framework.
- Diversified application scenarios: Demonstrates fragment linking and pharmacophore-guided design, suggesting practical utility beyond unconditional generation.
- Technical contribution: Clear integration of chemistry-aware constraints and templated reaction modeling that improve validity and retrosynthesis success. SYNSPACE is substantial, pairing reaction-level structure with optimized conformers, and includes pharmacophore annotations to support structure-informed tasks.

**Weaknesses:**

- Fixed reaction vocabulary: Reliance on fixed building blocks and templates restricts chemical diversity and may limit generalization performance.
- Baseline comparability: Many 3D baselines reported are not synthesis-aware. Stronger comparisons with synthesis-aware methods would be more convincing.
- Hard edge constraints can preclude valid chemotypes such as macrocycles.

**Questions:**

1. SYNCOGEN shows lower diversity and novelty. Is this mainly due to using fixed 93 building blocks and 19 reaction templates? How were these number chosen?
2. Can the model generate molecules with more building blocks than seen in training? How does performance changes for larger N?
3. For a fairer baseline, it would be more convincing to compare with other synthesis-aware models, such as CGFlow.
4. In SAMPLEEDGES, the authors draw parents stochastically. Did the authors test deterministic sampling for example argmax? How does this affect diversity and validity?

---

> ### Author Response · Authors · 2025-11-24
> **Response - reviewer 8SMJ (1/2)**
>
> We thank the reviewer for their feedback and suggestions. We are glad the reviewer highlighted our “well-motivated co-generation,” “practical utility beyond unconditional generation,” and “clear integration of chemistry-aware constraints.” Below we address each of the reviewer’s concerns:
>
> ### Weakness 1 and Question 1.
> We agree that our design space is more structured than fully atom-based models, and this is intentional. Our goal is a **synthetically realistic 3D design space that can be executed quickly in the lab**.
>
> Our building blocks and reactions were selected to satisfy favorable chemistry priors and wet-lab convenience: (i) inexpensive and readily available; (ii) empirically robust across many substrates; and (iii) mutually compatible under some standard reaction conditions. In contrast, many template libraries are (a) far too large to be physically stocked and (b) only nominally compatible (the mere presence of a functional group does not guarantee reactivity given steric/electronic context). The model itself is vocabulary-agnostic.
> We acknowledge that this pragmatic vocabulary contributes to slightly lower novelty/diversity metrics. However, the revised paper shows that the induced chemical space is still very broad:
> In the new SynSpace analysis compared to GEOM-Drugs (Appendix A.3), we find SynSpace displays impressive diversity and coverage of chemical space (**>400k Murcko scaffolds compared to ~90k in GEOM-Drugs**, in a more lead-like region (heavier, more polar, more sp³), all while allowing for easier in-house chemical synthesis.
> In the analysis of hard-to-synthesize structures (Appendix D.6), SynCoGen produces **significantly fewer problematic motifs**. This shows the curated design space is not only smaller, but better aligned with what can actually be made.
> As our experiments suggest, training on SynSpace yields competitive results to baseline methods, highlighting that the **vocabulary does not limit performance** (even on OOD conditional generation tasks, see Section 5.3).
>
> Nevertheless, we agree that the potential search space can be enlarged by training on even larger chemical spaces, and we are currently experimenting with a larger set of building blocks and reactions. We also commit to releasing a script that lets users regenerate SynSpace-style datasets from custom block/template sets.
>
> ### Weakness 2 and Question 3
> We agree that comparisons to synthesis-aware models are important. In the revision we add a **new pharmacophore-conditioned benchmark on 10 targets** (3 original + 7 from LIT-PCBA) with the following baselines: SynFormer (2D), CGFlow (2D+3D), and ShEPhERD (3D), please see Overall Response, “Stronger baselines” for details. Overall, we find SynCoGen achieves the highest retrosynthesis solve rates, the lowest incidence of hard motifs, diverse molecules, and better docking scores on 8/10 targets for all existing methods.
>
> ### Weakness 3
> The reviewer is correct. While inference-time edge constraints exclude macrocycles, we find this limitation acceptable in our setting. **Macrocycles are often challenging to synthesize at scale**, and as such do not quite align with our design objective of focusing on easily synthetically accessible building-block assemblies. Other chemotypes are available with the appropriate building blocks. We now clarify this point in the main text.

---

> ### Author Response · Authors · 2025-11-24
> **Response - reviewer 8SMJ (2/2)**
>
> ### Question 2
> Due to a lack of paddings, the current trained model cannot do so; however, __limiting generation to 5 building blocks already results in molecules that are close to the upper limit of molecular weight for drug-likeness__.
>
>
> | # BBs         | Avg. molecular weight (Da)      | Median MMFF94 energy (kcal/mol per atom) ↓ |
> |---------------|----------------------------------|---------------------------------------------|
> | SynSpace-3BB  | $358.67 \pm 71.30$               | $2.4831$                                     |
> | SynSpace-4BB  | $458.98 \pm 77.36$               | $2.3228$                                     |
> | SynSpace-5BB  | $565.52 \pm 83.80$               | $2.6040$                                     |
>
>
> Conventionally, most drug design pipelines filter for molecules under 550 da. Note that we see that the median energy does not experience a significant decline for larger molecules.
>
> We also note that most other molecule generators also impose size limits (typically in the form of heavy atom count) on their molecules to regulate generation and ensure that no molecule is too large to bind or absorb well. For instance, CGFlow limits generation to up to 3 building blocks. Nevertheless, we note that __architecturally, nothing prevents larger n__. Clarification is now added to App. B.10.
>
> ### Question 4
> We have added a **direct comparison between stochastic parent sampling and argmax selection** in App. D.2 using the same random seeds and sampling 1000 molecules.
>
> | Sampling Mode | Validity ↑ | PB_Valid ↑ | Diversity (intdiv) ↑ | Novelty (%) ↑ |
> |---------------|------------|------------|------------------------|----------------|
> | **Default**   | 0.947      | 0.8425     | 0.7811                 | 95.24          |
> | **Argmax**    | 0.956      | 0.8483     | 0.7797                 | 93.41          |
>
>
> We see that the proportion of valid molecules is slightly higher in the Argmax setting, at the cost of slightly lower diversity and novelty. However, these differences are relatively insignificant and performance is generally comparable across both settings.
>
> ### Concluding remarks
> We thank the reviewer again for their time and constructive feedback. We hope that the additional results and clarifications provided in the rebuttal address the concerns raised. If so, we would be encouraged if the reviewer could consider a fresher evaluation of our work in light of this new information. We are of course happy to provide any further details or analyses that might be helpful.

---

> ### Comment · Reviewer_8SMJ · 2025-11-28
>
> Thanks to the authors for the detailed reply. The revisions and additional experiments adequately resolve my earlier questions. I will keep my positive score.

---

### Author Response · Authors · 2025-11-24
**Overall response**

## 1/2

We thank all four reviewers for their careful and constructive feedback that allowed us to strengthen our paper. We are encouraged that all four reviews recognize (i) the importance of jointly modeling synthesizability and 3D structure, (ii) the technical soundness of the approach, and (iii) the practical relevance of the conditional tasks (fragment linking and pharmacophore-guided design), e.g., “well-motivated co-generation”, “sophisticated and well-suited generative paradigm”, and “results are convincing and demonstrate clear improvements over baselines”. In this global response, we address the shared clarification points across reviewers.

## Stronger baselines and expanded pharmacophore experiments (reviewers 8SMJ, 7GM3, 1Mne, t7Yp)

We have extended the pharmacophore-conditioned experiments from 3 to **10 protein targets** by adding 7 targets from LIT-PCBA. For each target, we now compare SynCoGen against
1. CGFlow (Shen et al., 2025): a recent 2D+3D synthesis aware model. We use their pocket-conditioned reward in a single amortized run [1] (rather than per-target retraining) to match our zero-shot amortized setting and avoid reward hacking.
2. SynFormer: a 2D synthesis-aware analogue generator. For a fair comparison, we condition on the native PDB ligand and evaluate the synthesizable projection task.
3. ShEPhERD: a 3D pharmacophore-conditioned generator.

These choices directly address the requests to compare against Shen et al. (2025) and other synthesis-aware methods, while keeping the evaluation setting as close as possible across methods (zero-shot, amortized generation, no per-target retraining). We note that other synthesis-aware methods (e.g., SynFlowNet and RGFN) require retraining from scratch per target using a binding-affinity reward function.

For each method, we generate 100 molecules for each of the 10 targets and dock them with AutoDock Vina (see revised Figure 5 and Appendices C.2 and D.6 for details and additional results). In terms of docking, SynCoGen **consistently generates molecules with superior or competitive docking scores relative to all other methods**. For synthesizability, we show that the curated set of building blocks and reactions used by SynCoGen yields designs that lie in **more synthesizable** regions of chemical space than SynFormer and CGFlow, as assessed by AiZynthFinder and Syntheseus, as well as by our breakdown of hard-to-synthesize chemical features in Appendix D.6 (Table 10). Comparing 3D vs 2D, SynCoGen and other 3D-informed models achieve better affinity scores than SynFormer, which operates in 2D, highlighting the **advantage of explicitly modeling molecular poses**. Summary metrics are shown below (docking results in Figure 5):
| Model     | Val. ↑ | AiZyn. ↑ | Synth. ↑ | PoseBusters ↑  | Div. ↑ |
|-----------|--------|----------|----------|-------|--------|
| SynFormer | 100.0  | 34       | 42       | --    | 0.82   |
| ShEPhERD  | 38.5   | 14       | 12       | 0.34  | 0.86   |
| CGFlow-ZS    | 100.0  | 45       | 16       | --    | 0.75   |
| SynCoGen  | 86.3   | 61       | 78       | 0.59  | 0.80   |



## Method clarifications, constraints, and ablations (reviewers 1Mne, t7Yp)

We thank the reviewers for pointing out that the method description can be convoluted and, at times, difficult to follow. To address this, we substantially **reorganize the exposition** of coordinate visibility masking (Section 4.2) and supplement the descriptions of compatibility logit masking (Section 4.3.3) and inference-time edge constraints (Section 4.4) **with diagrams and intuitive explanations** (see Appendices B.1, B.2, and B.3, respectively). We believe these diagrams help clarify the interplay of the various components that contribute to SynCoGen’s training and sampling process.

We also thank the reviewers for requesting ablations to justify these design choices. **Appendix D.1 contains explicit ablation results.** We clarify that the “Constraints” ablation includes both compatibility masking (restricting edges to reaction-feasible pairs) and sampling-time edge masking (enforcing tree-like, macrocycle-free reaction graphs). When these constraints are removed, model performance degrades significantly. We have revised Appendix D.1 to spell out precisely which components are toggled in each ablation setting and to highlight the impact of constraints.

---

> ### Author Response · Authors · 2025-11-24
> **Overall response**
>
> ## 2/2
>
> ## Search space and dataset analysis (reviewers 8SMJ, 7GM3, t7Yp)
> We report summary statistics for SynSpace compared to GEOM-Drugs in the table below (see Appendix A.3 for details). We find the resulting molecules to be **reasonable, diverse, and plausibly drug-like**. Compared to GEOM-Drugs, SynSpace molecules are somewhat more polar, with higher sp³ fraction and more rotatable bonds, which is typical of lead-like / late lead optimization compounds. Despite the restricted vocabulary, **SynSpace contains ~4.8× more unique Murcko scaffolds than GEOM-Drugs**, demonstrating that the combinatorics of the building blocks and reactions still yield a broad scaffold space.
>
> In the revised manuscript, we emphasize that these building blocks and reactions are chosen because they are **readily available, robust and known to undergo the selected reactions**(note that the mere presence of a nominally compatible group does not guarantee participation in the corresponding reaction, which is partially reflected in the lower synthesizability observed in the CGFlow/SynFormer experiments). In addition, we chose a smaller, curated set of building blocks to **facilitate cost-efficient in-house parallel synthesis**, since far fewer unique precursors must be purchased.
>
>
> | Property                                    | SynSpace | GEOM Drugs |
> |---------------------------------------------|----------|------------|
> | Molecular Weight                            | 492.16   | 355.83     |
> | Number of Heavy Atoms                       | 33.74    | 24.86      |
> | Octanol–Water Partition Coefficient (Log P) | 2.44     | 2.91       |
> | Number of Hydrogen Bond Donors              | 2.75     | 1.19       |
> | Number of Hydrogen Bond Acceptors           | 6.74     | 4.83       |
> | Quantitative Estimate of Drug-likeness      | 0.43     | 0.65       |
> | Fraction of sp³ Carbons                     | 0.41     | 0.30       |
> | Topological Polar Surface Area              | 111.32   | 73.73      |
> | Number of Rotatable Bonds                   | 6.95     | 4.90       |
> | Murcko Scaffold Number                      | 443,458  | 92,955     |
> | SAScore                                     | 3.34     | 2.51       |
>
>
> To evaluate how different models explore chemical space, we added an analysis (Appendix D.6) of undesirable substructures for the pharmacophore-conditioned generation experiments. The table below summarizes the fraction of generated molecules containing problematic motifs. This demonstrates that our building-block design effectively constrains the model to synthetically accessible regions of chemical space __without requiring additional filtering or manual post-processing__.
>
> | Model       | Macrocycles (%) ↓ | Fused Extreme (%) ↓ | Exotic Rings (%) ↓ | Hydrazine (%) ↓ | Peroxide (%) ↓ | Undesirable (any) (%) ↓ |
> |-------------|--------------------|----------------------|---------------------|------------------|-----------------|--------------------------|
> | CGFlow-ZS      | 0.0                | 31.1                 | 1.2                 | 0.0              | 0.0             | 31.3                     |
> | ShEPhERD    | 4.7                | 39.2                 | 31.2                | 0.0              | 0.3             | 46.8                     |
> | Synformer   | 1.8                | 1.9                  | 4.0                 | 1.3              | 0.0             | 8.3                      |
> | SynCoGen    | 0.0                | 0.1                  | 0.0                 | 0.0              | 0.0             | 0.1                      |
>
>
> While the current search space already covers over $10^9$ compounds, we agree that it is important for molecular generation models to scale to larger building-block sets. We note that the architecture allows for arbitrarily large building-block libraries; only the input and output layers of the model change in size as the vocabulary expands. We commit to releasing a simple command that constructs new datasets based on custom building blocks and reactions. We are currently curating and experimenting with an expanded library (e.g., 378 building blocks and 26 reactions) to explore even larger portions of chemical space, and we view this as a natural extension rather than a conceptual barrier. Future experiments will prioritize scaling the building-block library further.
>
> —
> References
>
> [1] Shen, T., Seo, S., Lee, G., Pandey, M., Smith, J. R., Cherkasov, A., Kim, W. Y., & Ester, M. (2024). TacoGFN: Target-conditioned GFlowNet for Structure-based Drug Design. arXiv preprint arXiv:2310.03223.

---

### Author Response · Authors · 2025-12-03
**Second Overall Response: Scalability with SynSpace-L**

We thank all reviewers again for their constructive comments. Following the rebuttal, we have conducted a substantial new experiment aimed at addressing the remaining concerns about the __fixed reaction vocabulary/scalability to larger chemical spaces__ (Reviewers 8SMJ, 7GM3, t7Yp). Reviewer 8SMJ has already indicated that our initial analysis and clarifications “adequately resolve my earlier questions”; here we show that SynCoGen scales empirically to a much larger, more diverse synthesis space in the __trillions__ with little loss in quality and improved synthesizability metrics. Note that we estimate the combinatorial size of the reachable molecular state space by computing the average reaction branching factor directly from the building-block–reaction compatibilities and propagating it multiplicatively across synthesis depth.

### SynSpace-L: 1000× larger virtual synthesis space
We extend SynSpace with a new dataset, **SynSpace-L**, constructed with the same principles as the original (commercially available, robust, mutually compatible), but with a markedly broader vocabulary:

- __378 building blocks__ and __26 reaction templates__ (vs. 93 and 19 originally),
- __~500k synthesizable molecular graphs and 4.2M conformers__,
- An estimated __search space of ~10¹³ molecules__ (1000x larger than our original set)
- __330k Murcko scaffolds__ (3.5x larger than Geom Drugs) with similar physicochemical profiles to the original SynSpace.

We then trained a new unconditional SynCoGen model on SynSpace-L for 50 epochs (short of full convergence, given the ~1000× larger combinatorial space). Even under this conservative training budget, the model retains high RDKit validity, realistic conformer energies, strong retrosynthesis solve rates, and __substantially higher novelty__ (99.1%).

| Method            | Valid ↑ | AiZyn ↑ | Synth ↑ | GFN-FF ↓ | GFN2-xTB ↓ | PB ↑ | Div ↑ | Nov ↑ |
|-------------------|---------|---------|----------|-----------|-------------|-------|--------|--------|
| SynCoGen-Original | 93.5    | 42      | 68       | 4.870     | -0.91       | 82.8  | 0.78   | 94.2   |
| SynCoGen-Large    |   87.0      |     52    |    77      |   5.502        |  -0.81           |   65.0    |    0.79    |    99.1    |
These results directly substantiate our earlier claims:
- SynCoGen is **vocabulary agnostic** and can be readily scaled to broader chemical spaces
- The original SynSpace was deliberately curated for lab executability and cost-efficient stocking.
- The enlarged vocabulary increases scaffold coverage, novelty, and synthesizability metrics simultaneously. This directly addresses the concern that a fixed, curated vocabulary must necessarily restrict diversity or bias the model too heavily.
The manuscript has been updated to incorporate SynSpace-L and the above analysis (highlighted in pink). We hope that these concrete scaling experiments reassure the AC and reviewers that SynCoGen’s vocabulary in the original submission is a __design choice for practicality__, and that the method extends naturally to substantially larger and richer synthesis spaces.

---

### Meta-Review · Area_Chair_6Rdx · 2026-01-06

**Summary:**

This paper presents SynCoGen, a framework that jointly generates synthesizable molecular reaction graphs and 3D coordinates using masked discrete diffusion and flow matching. The authors also propose SynSpace, a dataset of 1.2M synthesis-aware molecules with 7.5M conformers. The method demonstrates strong performance on unconditional generation, fragment linking, and pharmacophore-conditioned tasks.

**Reviewer Concerns:**

Addressed concerns:
- Lack of baselines (8SMJ, 7GM3, 1Mne)
- Method clarity (1Mne, t7Yp)
- Dataset characterization (7GM3)

Partially addressed:
- Scalability to larger vocabularies (8SMJ, 7GM3, t7Yp): Authors propose new SynSpace-L experiments, but this was trained for only 50 epochs and results show some degradation in PoseBusters validity.

**Reviewer Scores:**

Most of the concerns were resolved and the reviewers would have raised their scores.

---

### Decision · Program_Chairs · 2026-01-26

Accept (Poster)